# Memory CD4 T cell subsets are kinetically heterogeneous and replenished from naive T cells at high levels

Graeme Gossel[1,2†§], Thea Hogan[3†], Daniel Cownden[1], Benedict Seddon[3*‡], Andrew J Yates[1*‡]

[1]Institute of Infection, Immunity and Inflammation, College of Medical, Veterinary & Life Sciences University of Glasgow, Glasgow, United Kingdom; [2]Department of Genetics and Genomic Sciences, Icahn School of Medicine at Mount Sinai, New York, United States; [3]Institute of Immunity and Transplantation, University College London, London, United Kingdom

**\*For correspondence:** benedict. seddon@ucl.ac.uk (BS); andrew. yates@glasgow.ac.uk (AJY)

[†]These authors contributed equally to this work
[‡]These authors also contributed equally to this work

**Present address:** [§]Department of Physics and Astronomy, Hunter College, City University of New York, New York, United States

**Competing interests:** The authors declare that no competing interests exist.

**Abstract** Characterising the longevity of immunological memory requires establishing the rules underlying the renewal and death of peripheral T cells. However, we lack knowledge of the population structure and how self-renewal and de novo influx contribute to the maintenance of memory compartments. Here, we characterise the kinetics and structure of murine CD4 T cell memory subsets by measuring the rates of influx of new cells and using detailed timecourses of DNA labelling that also distinguish the behaviour of recently divided and quiescent cells. We find that both effector and central memory CD4 T cells comprise subpopulations with highly divergent rates of turnover, and show that inflows of new cells sourced from the naive pool strongly impact estimates of memory cell lifetimes and division rates. We also demonstrate that the maintenance of CD4 T cell memory subsets in healthy mice is unexpectedly and strikingly reliant on this replenishment.

## Introduction

The diversity and longevity of T cell memory are shaped by influx, cell division and cell death. A major challenge is to understand how these processes are regulated in health and how they respond to perturbations such as chronic infections. To understand the evolution of immune repertoires within a host therefore requires quantifying homeostatic processes, identifying the rules of replacement within memory subsets, and characterising any distinct homeostatic niches that lie within them. The dominant approaches to studying T cell population dynamics at steady state involve the adoptive transfer of cells labelled with inert dyes such as CFSE and using its rate of dilution to infer rates of proliferation (*De Boer et al., 2006*; *Choo et al., 2010*), or tracking the uptake and dilution of labels incorporated into the DNA of dividing cells, the kinetics of which contain information regarding both division and loss of cells (*De Boer and Perelson, 2013*). In both cases, mathematical models are needed to interpret the data. However, while for some T cell subsets in mice and humans there is broad agreement regarding basic parameters such as population-averaged cell lifetimes, discrepancies remain and defining homeostatic dynamics in detail is difficult. Gaps in our understanding of how memory compartments are structured, and how the processes of cell division and death are correlated, generate uncertainty in how to formulate the dynamical models to describe data from labelling studies (*De Boer et al., 2003*) and indeed these data may not be sufficiently rich in information to allow us to discriminate between these models (*De Boer et al., 2012*).

**eLife digest** The immune system protects the body from the infectious bacteria, viruses and other microorganisms present in our everyday environment (collectively known as pathogens). One feature of this system is that it can form long-lasting memories of the pathogens it has previously encountered by creating cells called memory cells. When the same pathogen invades the body again, the memory cells help the immune system to kill off the infection more rapidly and efficiently than before. This process also underlies how vaccines work. By exposing the immune system to a pathogen in a controlled, safe way, memory cells form that can efficiently fight off a future infection.

Do immune memories only form when we are sick with infections? Or does constant exposure to the microbes that are present in the natural environment also stimulate the formation of memory cells? Also, how does the formation of new memory cells affect the existing memory cells?

To answer these questions, Gossel, Hogan et al. studied laboratory mice that were kept in a clean, controlled environment – and not exposed to pathogens – for a year. This timespan represents about half of a mouse's normal lifespan. Over the course of the year, new immune memory cells constantly formed in the mice. Furthermore, in young healthy mice up to a tenth of the existing immune memory cells were replaced each week. Despite the constant formation of new memory cells, the overall number of immune memory cells in the mice only doubled over the course of the year, suggesting that some memory cells must also be lost.

The discovery that new immune memory cells are constantly made raises new questions to be investigated in future studies. For example, does the constant formation of memory cells make it harder to retain useful memories of pathogens, and does this explain the need for booster vaccinations?

One major difficulty is measuring the contribution that any influx of new cells originating from naive precursors makes to the maintenance of immune memory. While it is clear that new memory T cells are generated during infections and by the seeding of empty peripheral compartments, either early in ontogeny (*Le Campion et al., 2002*) or following reconstitution in irradiation chimeras (*Surh and Sprent, 2008*), it is unclear whether there continue to be significant contributions from the naive pool in the steady state in normal healthy hosts, in the absence of infection. Labelling experiments can be inconclusive in this regard. In early studies, flows into cell populations of interest – for example, entry of recent thymic emigrants into the mature naive T cell pools, or cells moving from the naive pools into memory through antigen-driven expansion – were invoked to explain observations that average rates of division and death were estimated to be unequal, despite the populations being apparently stable in size (*Mohri et al., 1998, 2001*; *Bonhoeffer et al., 2000*). However, in some cases the nature of these source terms was puzzling. First, the required magnitude of the source in studies of naive T cell turnover vastly exceeded what was expected from the thymus (*Asquith et al., 2002*; *Borghans and de Boer, 2007*). Second, sources were also required to be largely unlabelled during label administration (*Mohri et al., 2001*; *Asquith et al., 2002*), but memory precursors might be expected to have divided – and thus incorporated the label – shortly before entering the memory pool.

An alternative explanation of these observations reflects heterogeneity within T cell compartments. A key study (*Asquith et al., 2002*) pointed out that differences in division and death rates inferred from the accrual and loss of label need not derive from influx and instead may arise if the behaviour of labelled (and so recently-divided) cells does not reflect the population average. Such heterogeneity can be classified into two non-exclusive types; kinetic (*Asquith et al., 2002*; *Macallan et al., 2003*; *Asquith et al., 2006*; *Borghans and de Boer, 2007*; *Vrisekoop et al., 2008*; *Asquith et al., 2009*; *Ganusov et al., 2010*; *Ganusov and De Boer, 2013*; *De Boer et al., 2012*; *De Boer and Perelson, 2013*; *Westera et al., 2013*) and temporal (*Grossman et al., 1999*; *Bonhoeffer et al., 2000*; *Ribeiro et al., 2002*; *De Boer et al., 2003, 2012*; *De Boer and Perelson, 2013*). A kinetically heterogeneous population comprises two or more subpopulations with different rates of division and/or death, and short-term labelling experiments will tend to over-sample those dividing fastest. Temporal heterogeneity reflects the idea that cells within a single population may

display different rates of division or turnover (loss) at different times. For example, quiescent and dividing or recently-divided cells may be differentially susceptible to death. This form of heterogeneity has been invoked in models of T cell clonal expansion (*Bonhoeffer et al., 2000*; *Ribeiro et al., 2002*; *De Boer et al., 2003*, *2012*) and homeostasis (*De Boer et al., 2012*). Discriminating between kinetic and temporal heterogeneity with DNA labelling alone is challenging (*De Boer et al., 2012*; *De Boer and Perelson, 2013*), but doing so has the potential to give us mechanistic insights into T cell homeostasis on different levels. Kinetic heterogeneity likely reflects phenotypic substructure, and suggests the existence of distinct homeostatic or ecological niches if these subpopulations are stably maintained. In contrast, the extent of temporal heterogeneity – that is, how division and death are coupled – provides clues as to how numbers are regulated at the single-cell level.

In short, our understanding of memory T cell homeostasis is limited because the effects of external sources of cells and heterogeneity in population dynamics may mimic or mask one another and it is difficult to distinguish them with conventional approaches. In this study we aimed to disentangle these processes, focusing on memory CD4 T cell subsets in mice. We use a temporal fate mapping method to directly estimate the constitutive rates of flow of cells into different memory CD4 T cell compartments. We then generate fine-grained timecourses of DNA labelling combined with measurements of cell cycle status, a strategy that when paired with the independent estimates of memory influxes yields sufficient information to discriminate between alternative models of population dynamics. This combined approach allows us to estimate for the first time the contributions at steady state of de novo production of memory cells and production through division of existing memory cells, as well as yielding insights into the cellular mechanisms regulating memory CD4 T cell subsets in mice.

## Results

### Naive T cells transition to memory in the steady state and in the absence of deliberate infection

It is an immunological paradigm that activation of naive T cells by foreign antigens ultimately gives rise to persistent populations of memory cells. However, in healthy individuals, not deliberately infected, it is unclear whether generation of memory is an event restricted to first encounter with environmental antigens, such as in the establishment of the T cell compartment in neonates, or whether generation of memory cells occurs constitutively throughout life. Whether there is continual differentiation of naive cells into memory in the steady state has not previously been assessed, and knowledge of this quantity is critical for quantitative analysis of memory homeostasis. To characterise the fluxes into memory subsets, we took advantage of a temporal fate mapping approach described previously (*Hogan et al., 2015*) that allows visualisation of tonic reconstitution processes within different haematopoetic compartments. Briefly, we condition young adult CD45.1 hosts with the chemotherapeutic drug busulfan that ablates haematopoetic stem cells (HSC) but leaves compartments of committed lineages intact, including thymic and peripheral T cells. Conditioned hosts are then reconstituted with CD45.2 bone marrow (*Figure 1A*). Total numbers of thymocytes remain normal, and by 6 weeks the CD45.1:CD45.2 ratio equilibrates in all thymic compartments (*Hogan et al., 2015*). We see no trend in thymic chimerism across treated animals out to a year post-BMT (*Hogan et al., 2015*), indicating that chimerism among T cell precursors is stably maintained. Chimeric mice also exhibit normal numbers of peripheral CD4 naive and CD4 memory T cells (*Figure 1B*), and both populations display normal levels of proliferation as assessed by Ki67 expression (*Figure 1C*). Together these data indicate that busulfan treatment and the generation of bone marrow chimerism have no meaningful impact on lymphocyte homeostasis, as previously described (*Vezys et al., 2006*; *Hogan et al., 2015*). By 8 weeks post-BMT, donor-derived cells are readily detectable not only in the naive but also to a striking extent in the CD44[hi] memory compartment (*Figure 1D*), revealing that well into adulthood newly generated naive T cells continue to differentiate into memory in clean healthy mice. We observe a steady but ultimately incomplete replacement of the host-derived CD4 memory cells with donor cells over the course of a year (*Figure 1E*), while the total CD4 memory compartment remains relatively stable in size (*Figure 1B*).

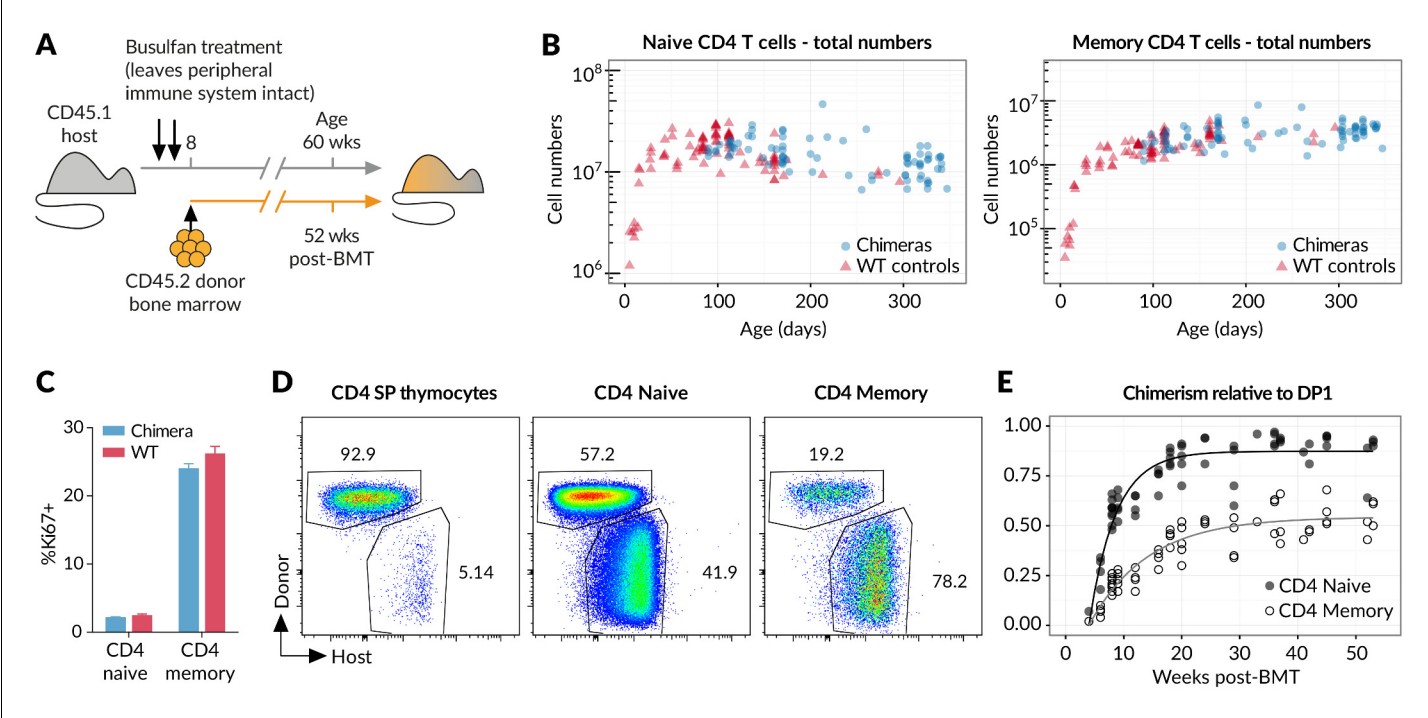

**Figure 1.** New donor T cells differentiate into memory compartments in the absence of deliberate infection. (**A**) Outline of experimental protocol. Host CD45.1 mice aged 8 weeks were treated with two doses of 10 mg/kg busulfan, followed by injection of $10^7$ T cell-depleted bone marrow cells from CD45.2 donors. The numbers of donor and host cells in the thymus and peripheral lymphocyte compartments were evaluated by flow cytometry at various time points up to one year post bone marrow transplantation (BMT). (**B**) Numbers of naive and memory CD4 T cells (host + donor) recovered from spleen and lymph nodes of busulfan chimeras made at age 8 weeks, compared to numbers in WT CD45.1 controls. (**C**) Ki67 expression in naive and memory CD4 cells in chimeras (14 weeks post-BMT) compared to age-matched WT controls; 11 mice per group. (**D**) Identification of host and donor-derived cells in a representative mouse 8 weeks post-BMT. (**E**) Timecourses of normalised peripheral chimerism (defined as the proportion of the population that is donor-derived, divided by the proportion of the DP1 population that is donor-derived) in naive and memory CD4 T cell populations, showing steady but incomplete replacement of host cells in both. Fitted curves are empirically determined to show trends only.

The following source data is available for figure 1:

**Source data 1.** Comparing naive and memory cell numbers and Ki67 expression in busulfan chimeras and wild-type controls (panels B and C) .
**Source data 2.** Timecourses of infiltration of donor-derived T cells into the naive and memory compartments in busulfan chimeras (panel E).

## Measuring the constitutive flows into CD4 T cell memory subsets

Since the continuous generation of new memory T cells was readily detectable, we examined the underlying dynamics more closely, paying particular attention to the feeding of canonically defined memory subsets. Specifically, we modelled the kinetics of replacement of host cells by donor cells within the $CD4^+CD25^-CD44^{hi}CD62L^{hi}$ and $CD4^+CD25^-CD44^{hi}CD62L^{lo}$ populations, termed CD4 central memory (CD4 $T_{CM}$) and CD4 effector memory (CD4 $T_{EM}$) respectively (*Figure 2A*), for over a year post-treatment. We previously demonstrated that the kinetics of lymphocyte replacement in the busulfan chimeras are a rich source of information regarding homeostatic turnover and population substructure (*Hogan et al., 2015*).

Assuming host and donor cells behave similarly, the rate of accumulation of donor cells in each subset is a constant fraction of the total rate of influx of cells from the naive pool, presumably following clonal expansion, and/or through differentiation from other memory subsets. We used simple mathematical models to describe these flows (see Materials and methods). Our choice of models was guided by two key observations. First, the appearance of donor CD4 $T_{EM}$ cells lagged that of both naive and $T_{CM}$ cells (*Figure 2C* and *Figure 2—figure supplement 1*), suggesting that CD4 $T_{CM}$ are sourced predominantly from naive precursors while CD4 $T_{EM}$ may be sourced either directly

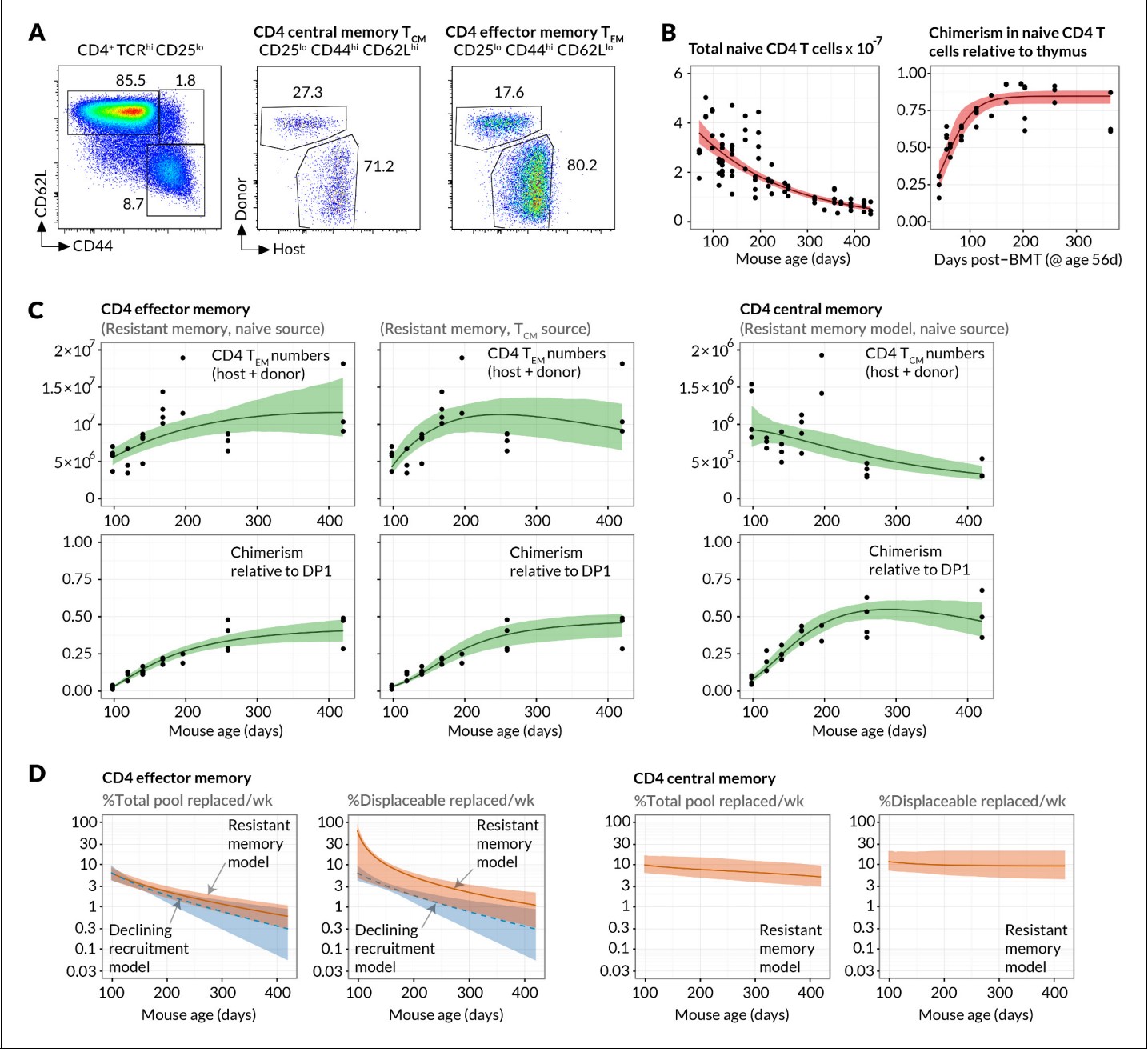

**Figure 2.** Estimating constitutive rates of generation of CD4 T cell memory. (**A**) Gating strategy for CD4 central and effector memory subsets. (**B**) Describing the kinetics of the source. Fits of empirical descriptor functions to the timecourses of naive CD4 counts and chimerism, with 95% uncertainty envelopes (see Materials and methods). Similar curves (not shown) were used to describe CD4 $T_{CM}$ numbers and chimerism when modelled as the source for CD4 $T_{EM}$. Estimates of the parameters defining the source functions are in **Appendix 1—table 1**. (**C**) Timecourses of total (host+donor) numbers of CD4 $T_{CM}$ and $T_{EM}$ and of chimerism, modelled from 6 weeks post-BMT (age 14 weeks/98 days). The resistant memory models with naive source described both the CD4 $T_{CM}$ and $T_{EM}$ data well (left-hand and central panels). Also shown are the statistically poorer fits to CD4 $T_{EM}$ kinetics using a model in which they are fed exclusively by CD4 $T_{CM}$ (ΔAIC = 11). Both models contained five free parameters; estimates are in **Appendix 1—table 2**. (**D**) Projections of how the rates of memory replacement change with age, assuming a naive source. Replacement is shown both as a fraction of the total pool, and as a fraction of the displaceable subset only.

The following source data and figure supplements are available for figure 2:

**Source data 1.** Timecourses of numbers and chimerism within the naive, effector memory and central memory CD4 T cell compartments in busulfan chimeras (**Figure 2** panels B and C, and **Figure 2—figure supplement 1**).

*Figure 2 continued on next page*

*Figure 2 continued*

**Source data 2.** Source code used to analyse flows between naive, CD4 T$_{EM}$ and CD4 T$_{CM}$ populations.
**Figure supplement 1.** Early kinetics of peripheral replacement in busulfan chimeras made at age 8 weeks, showing that the generation of CD4 T$_{EM}$ cells lags that of CD4 T$_{CM}$ (three mice per timepoint; mean and s.e.m.).
**Figure supplement 2.** Estimated sizes of memory populations resistant to displacement.

from the naive compartment and/or via CD4 T$_{CM}$. Second, donor cells displayed a more restricted capacity for populating memory relative to the naive pool (*Figure 1E*). We considered two explanations of this observation; either that there exist self-renewing populations of host-derived memory cells that resist displacement by newly recruited cells (the 'resistant memory' model), or that the per-cell rates of entry into each memory subset wane over time (the 'declining recruitment' model).

We then assessed the abilities of these models to describe the kinetics of the sizes and host/donor compositions of the CD4 T$_{EM}$ and CD4 T$_{CM}$ pools in healthy, chimeric laboratory mice aged between 14 and 60 weeks. For both memory subsets we fitted each combination of model and source population to the timecourses of the total numbers and donor chimerism of cells recovered from spleen and lymph nodes. The donor chimerism was normalised to that at the early DP1 stage of thymic development in each animal. Doing this controlled for varying degrees of depletion of HSC across animals with busulfan treatment. The size and donor/host composition of the putative source populations (naive for CD4 T$_{CM}$, and either naive or T$_{CM}$ for CD4 T$_{EM}$) were not modelled explicitly but instead described by empirical functions fitted to the observations (*Figure 2B*; Materials and methods). Together these steps allowed us to describe the data from multiple mice with single predictors reflecting the population-average parameters (*Figure 2C*; Materials and methods).

For both models explaining the apparent capping in host memory cell replacement, we compared the variant models in which CD4 T$_{EM}$ was fed either directly from naive or from CD4 T$_{CM}$. Comparing these fits, we found the strongest statistical support for a dominant naive CD4 → T$_{EM}$ recruitment pathway over CD4 T$_{CM}$ →T$_{EM}$ (ΔAIC = 11). Although the model fits were visually similar (*Figure 2C*), the conclusion favouring a naive source derives largely from substantial differences in the quality of the fits during the early stages of infiltration of donor cells into memory, which are relatively data-rich and well defined. Thus, naive T cell numbers, and not CD4 T$_{CM}$ numbers, provide the strongest predictor of CD4 T$_{EM}$ accumulation over long timescales.

For CD4 T$_{EM}$ we found comparable statistical support for the resistant memory and declining recruitment models (ΔAIC = 0.16). For CD4 T$_{CM}$ we found stronger support for the resistant memory model (ΔAIC = 8.3). We cannot rule out a combination of resistant memory and declining recruitment, and by parsimony we favour the resistant memory model for both populations (fits shown in *Figure 2C*). We estimate that in 14 week-old mice, 6.3% of CD4 T$_{EM}$ (95% confidence interval 4.4–8.6) and 9.8% (6.5–16.9) of CD4 T$_{CM}$ are displaced per week by new memory cells from the source. The declining recruitment model yielded comparable parameter estimates (*Table 1*). Indeed, consistent rates of production of new memory cells could be derived simply from the growth rate of chimerism in memory and the difference in chimerism between it and its source, irrespective of the details of the mechanism limiting memory replacement (see Materials and methods).

If this seeding of new memory occurs through recruitment of naive cells followed by clonal expansion and differentiation, one would expect there to be a delay in the transition between source and memory. To explore this we extended the models to allow for lags of 1–7 days before changes in the source population were reflected in changes in the rate of entry into memory. During this period the transitioning cells would likely disappear from the naive and memory T cell populations as we defined them (see Materials and methods), through expression of the IL-2 receptor $\alpha$-chain, CD25. These extensions yielded rates of replacement that were very similar to the zero-lag models, with weaker statistical support, although the timecourses lack the resolution required to examine this transition process in detail.

The models can also be used to predict how the rates of replacement of CD4 T$_{EM}$ and CD4 T$_{CM}$ change with age (*Figure 2D*), although these predictions derive from relatively uncertain projections of the sizes of the populations beyond a year of age (*Figure 2C*). We predict that between 14 weeks

**Table 1.** Estimated rates of replacement of CD4 effector and central memory through influx of new cells in 14 week-old mice. We quote both absolute influx (cells per day as a percentage of the pool size) and percent replaced per week. The latter is slightly less than 7 × the daily rate of influx because immigrant cells are assumed to be lost at the same rate as existing displaceable cells (see Methods). AIC differences (ΔAIC) are quoted relative to the best-fitting model for each cell type. These differences reflect the relative support for two models, with exp(−ΔAIC/2) being the relative probability that it is the model with the lower penalised likelihood (larger AIC value) that minimises the information lost in describing the data. For CD4 $T_{EM}$ the two models have equal support, but for CD4 $T_{CM}$ the resistant memory model is favoured (ΔAIC = 8.3, exp(−ΔAIC/2)=0.02). Models considered most plausible are highlighted.

| | | Resistant memory model | | | | | Declining recruitment model | | | | |
| --- | --- | --- | --- | --- | --- | --- | --- | --- | --- | --- | --- |
| | Source | % Input/day | | % Replaced/wk | | ΔAIC | % Input/day | | % Replaced/wk | | ΔAIC |
| CD4 $T_{EM}$ | Naive | 1.0 | (0.7, 1.4) | 6.3 | (4.4, 8.6) | 0.16 | 1.0 | (0.7, 1.6) | 6.4 | (4.3, 9.7) | 0 |
| | CM | 4.1 | (2.6, 7.5) | 23.0 | (16, 35) | 11 | 3.6 | (2.1, 6.2) | 21 | (13, 30) | 11 |
| CD4 $T_{CM}$ | Naive | 1.5 | (1.0, 2.8) | 9.8 | (6.5, 17) | 0 | 2.3 | (1.3, 4.1) | 13.5 | (8.4, 22) | 8.3 |

and 1 year of age, the proportion of cells replaced each week by new memory falls from 6.3% to 1% for CD4 $T_{EM}$ and 9.8% to 6% for CD4 $T_{CM}$. For CD4 $T_{EM}$ this decline stems from a combination of the fall in naive T cell numbers (the putative source population) with age, and a predicted slow increase in CD4 $T_{EM}$ numbers (*Figure 2C*). The declining recruitment model predicts a steeper drop in rates of replacement with age, due to the multiplicative effect of the fall in both naive T cell numbers and the *per capita* rate of recruitment from the naive pool with age (*Figure 2D*, blue shaded regions). For CD4 $T_{CM}$ the proportional replacement remains relatively steady with age, because the drop in the size of the naive source population is balanced by the predicted slow decline in CD4 $T_{CM}$ numbers.

Finally, we estimate that between 14 weeks and 1 year of age the resistant, numerically stable memory populations make up 16% to 40% of CD4 $T_{CM}$ and 96% to 46% of CD4 $T_{EM}$, though with some uncertainty (*Figure 2—figure supplement 2*). Throughout this period approximately 10% of the remaining displaceable CD4 $T_{CM}$ subpopulation is replaced each week. For CD4 $T_{EM}$, because the resistant population at 14 weeks of age is estimated to be a large proportion of the pool and the source is substantial, we predict that as much 65% of displaceable CD4 $T_{EM}$ are replaced per week. This rate falls to 1.5 %/week in year-old mice as the displaceable population grows and the rate of immigration falls in tandem with naive T cell numbers (*Figure 2D*, right-hand panels).

In summary, we find clear evidence for substantial tonic flows of cells from the naive T cell pool into both CD4 central and effector memory. For central memory we favour a model in which this flow remains high well into the second year of life, but displaces only a subset of cells. The remainder are generated before 8 weeks of age and analogous to the apparently stable 'incumbent' populations of naive CD4 and CD8 T cells that also resist replacement (*Figure 2B*, right-hand panel; and *Hogan et al. (2015)*). We estimate that CD4 effector memory is replaced at a rate comparable to that of central memory in young adult mice, but that the rate of assimilation of new effector memory cells declines more strongly with age. This kinetic can be explained equally well by the existence of a resistant CD4 $T_{EM}$ subset or simply by a waning force of recruitment from the naive pool.

## Using Ki67 expression as a molecular clock permits temporal stratification of DNA label uptake

Having identified and measured the contributions to CD4 memory subsets from naive sources, we wanted to measure cell lifetimes and division rates within these subsets in normal healthy mice and to test alternative models of homeostatic dynamics. Resolving different types of heterogeneity in these dynamics requires dissecting the fates of quiescent and dividing or recently-divided cells. Doing so is difficult with DNA labelling alone because for anything other than very short pulse-chase experiments the labelled fraction contains cells with a wide range of times since their last division. We therefore measured the division-linked uptake of the nucleoside analogue 5-bromo-2'-deoxyuridine (BrdU) in the context of Ki67 expression. Ki67 is a nuclear protein that is expressed during cell division but subsequently lost by non-dividing cells on a timescale of a few days (*Pitcher et al., 2002*; *Younes et al., 2011*; *De Boer and Perelson, 2013*). As such, it is a marker of active and

recent division. The frequency of cells expressing Ki67 is expected to be constant in a population at steady state, but when combined with time courses of BrdU labelling, Ki67 acts as a timestamp allowing us to distinguish the fates of recently divided Ki67$^{high}$ BrdU$^+$ cells and their quiescent Ki67$^{low}$ BrdU$^+$ progeny (*Figure 3A*).

We performed three pulse-chase experiments in which mice were fed BrdU for either 4, 7 or 21 days, with a chase period of 8–14 days following withdrawal of the label from drinking water. Groups of mice were analysed for co-staining of Ki67 and BrdU at different times during these experiments

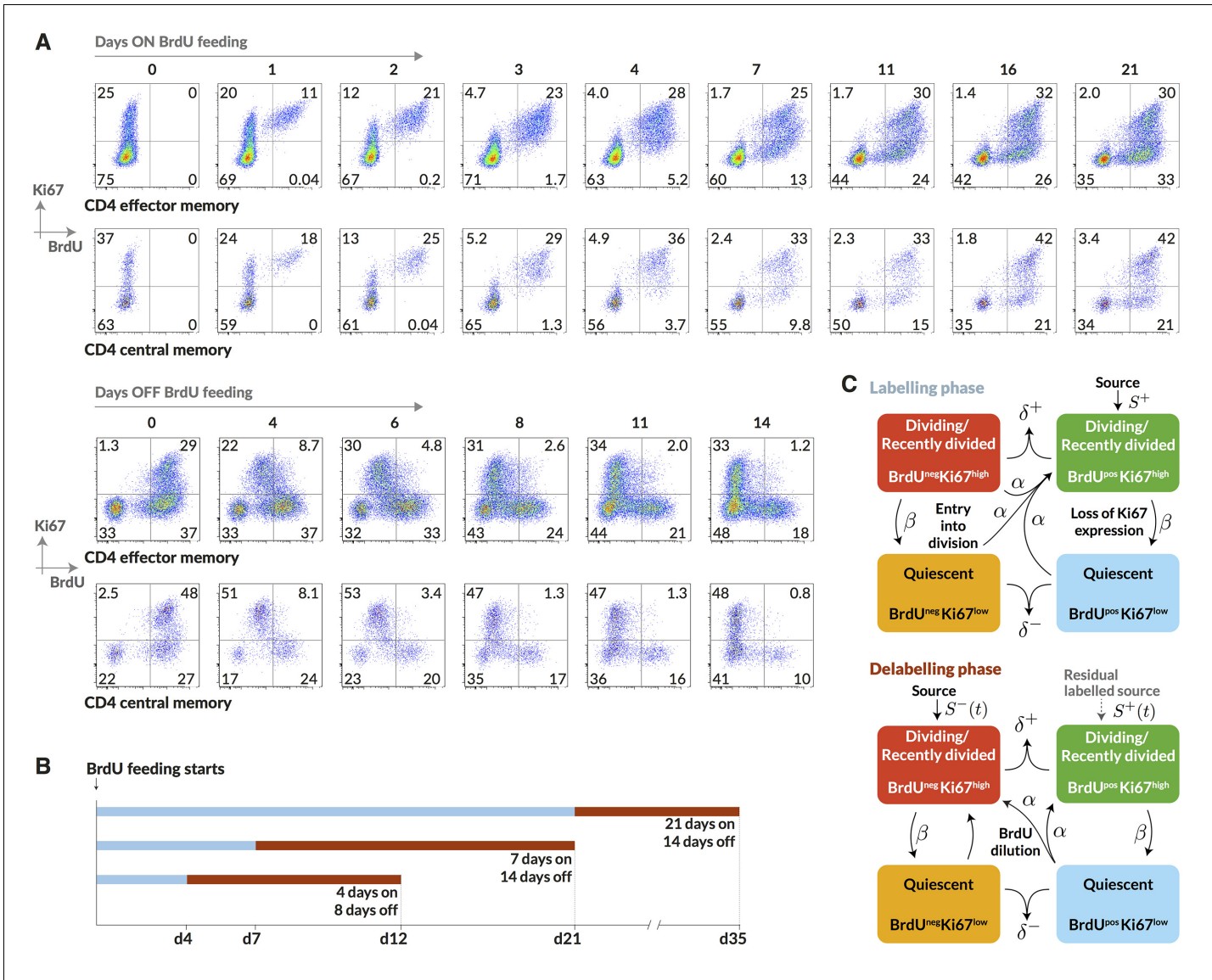

**Figure 3.** Quantifying the homeostatic dynamics of effector and memory CD4 T cells by combining BrdU labelling with measurements of Ki67 expression. (A) Representative data from flow cytometric analyses of BrdU uptake and Ki67 expression in a pulse-chase experiment. Cells were recovered from lymph nodes. (B) Outline of experimental design. (C) A schematic of the core multi-compartment model used to describe the flows between the BrdU$^{-/+}$ × Ki67$^{low/high}$ populations during and after labelling. Shown here is a model of temporal heterogeneity, in which either effector or central memory CD4 T cells are modelled as a single population entering division stochastically at *per capita* rate $\alpha$; with quiescent (Ki67$^{low}$) and recently divided (Ki67$^{high}$) cells dying at rates $\delta^-$ and $\delta^+$ respectively; an external source of cells feeding the BrdU$^\pm$ Ki67$^{high}$ populations at rates $S^+$ and $S^-$, where $S^+ + S^-$ is a constant, $S$; and cells transitioning from Ki67$^{high}$ to Ki67$^{low}$ at rate $\beta$. This basic model was refined to account for multiple subpopulations (kinetic heterogeneity), different distributions of Ki67 expression times, inefficient BrdU uptake, and post-labelling dilution of BrdU within both labelled cells and within the source ($S^{+/-}(t)$). See Materials and methods and Appendix 1 for details of the model formulation.

to define the population dynamics of CD4 $T_{EM}$ and CD4 $T_{CM}$ in detail (*Figure 3B*; for the experimental protocols see Materials and methods M1). Over the relatively short time courses of the experiments we saw no substantial changes in either the absolute sizes of memory T cell subsets or in the fraction of cells expressing Ki67 (*Appendix 1—figure 1*). With these constraints the dynamics of the system can be characterised by two quantities – the proportions of cells within the Ki67$^{high}$ and Ki67$^{low}$ populations that are BrdU$^+$. In this equilibrium, loss (turnover) of memory is balanced by production of new cells by division and input from external sources.

## Modelling BrdU/Ki67 kinetics reveals strong support for kinetic over pure temporal heterogeneity in CD4 T cell memory subsets

To assess the support for different homeostatic mechanisms, we used mathematical models to describe the fluxes of cells between the BrdU$^{+/-}$ × Ki67$^{high/low}$ populations within the CD4 $T_{EM}$ and CD4 $T_{CM}$ subsets (*Figure 3C* and Materials and methods; detailed in Appendix 1). In a model of pure temporal heterogeneity (TH), each memory subset is assumed to comprise one population of cells undergoing single stochastic divisions characteristic of T cell homeostasis (*Yates et al., 2008*; *Choo et al., 2010*; *Hogan et al., 2013*), but with potentially different rates of loss of quiescent (Ki67$^{low}$) and recently divided (Ki67$^{high}$) cells. We also considered a model of pure kinetic heterogeneity (KH) in which each memory subset is assumed to comprise two sub-populations maintained independently, each at constant size and with their own rates of division and loss, and with Ki67$^{high}$ and Ki67$^{low}$ cells within each subpopulation having equal susceptibility to death. In both TH and KH models, any external source is assumed to feed the Ki67$^{high}$ subpopulation(s) exclusively.

We aimed to estimate the rate(s) of division and loss in both models, together with a minimal set of additional parameters representing key biological quantities. In the KH models these included the relative sizes of the two subpopulations and the allocation of the source into each. Both TH and KH models also required parameters quantifying the efficiency of BrdU uptake per cell division and the gradual decline of the BrdU$^+$ fraction once BrdU feeding stops. The latter can result from three non-exclusive processes; (i) differences in the death rates of BrdU$^-$ and BrdU$^+$ cells, (ii) dilution of the labelled population by unlabelled cells from the source (*Tough and Sprent, 1994*; *Bonhoeffer et al., 2000*; *Debacq et al., 2002*; *De Boer et al., 2003*), and (iii) within-cell dilution of BrdU through division post-administration (*Tough and Sprent, 1994*; *Parretta et al., 2008*; *Ganusov et al., 2010*). The first process is captured in the basic KH/TH model structure. The second requires a description of the dilution of label within the source post-administration. We found the strongest support for a simple model in which the BrdU content of the souce drops rapidly from 100% to zero after a delay that is estimated from the data (see Appendix 1). For the third, we found that the best-fitting models required two divisions to drive cells from BrdU$^+ \rightarrow$ BrdU$^-$, consistent with another BrdU labelling study in mice (*Parretta et al., 2008*). Finally, we explored different distributions of times spent in the Ki67$^{high}$ state post-mitosis, by assuming cells progress through a variable number of intermediate states before transitioning to Ki67$^{low}$. Best-fitting models for both KH and TH require more than 12 such states, meaning that there is very little variance in the time cells spend in the Ki67$^{high}$ state. We note that these kinetics and the estimated mean residence time in Ki67$^{high}$ ($1/\beta$) reflect the Ki67 gating strategy as well as the cell-intrinsic rate of loss of Ki67 post-mitosis. Exploiting the constraints that there were no significant changes in the numbers and the proportions of cells that were Ki67$^{high}$ within both CD4 $T_{EM}$ and $T_{CM}$ during the labelling experiments (see *Appendix 1—figure 1*), four free parameters remained for the TH model and six for the KH model. A detailed description of the model formulation and the strategy for parameter estimation is given in Appendix 1.

Strikingly, despite this freedom in parameterisation, the data were sufficiently rich to discriminate between the models and showed unequivocal support for kinetic over pure temporal heterogeneity within the CD4 effector and central memory pools (*Figure 4*, ΔAIC = 110 (CD4 $T_{EM}$), 251 (CD4 $T_{CM}$)). For both memory subsets the BrdU/Ki67 timecourses were consistent with the existence of two subpopulations roughly equal in size but with highly distinct kinetics (*Table 2* and *Figure 5A*, at dashed vertical lines). CD4 $T_{CM}$ appear to comprise a population dividing and dying roughly every 3 days, and a slower population with mean lifetime of 38 days, dividing every 170 days, with the source feeding the slow and fast populations in roughly a 2:1 ratio. For CD4 $T_{EM}$ the fast population appears to be essentially self-renewing, dividing and dying every 6 days, with the slower population fed by the source (mean lifetime 43 days, interdivision time 140 days).

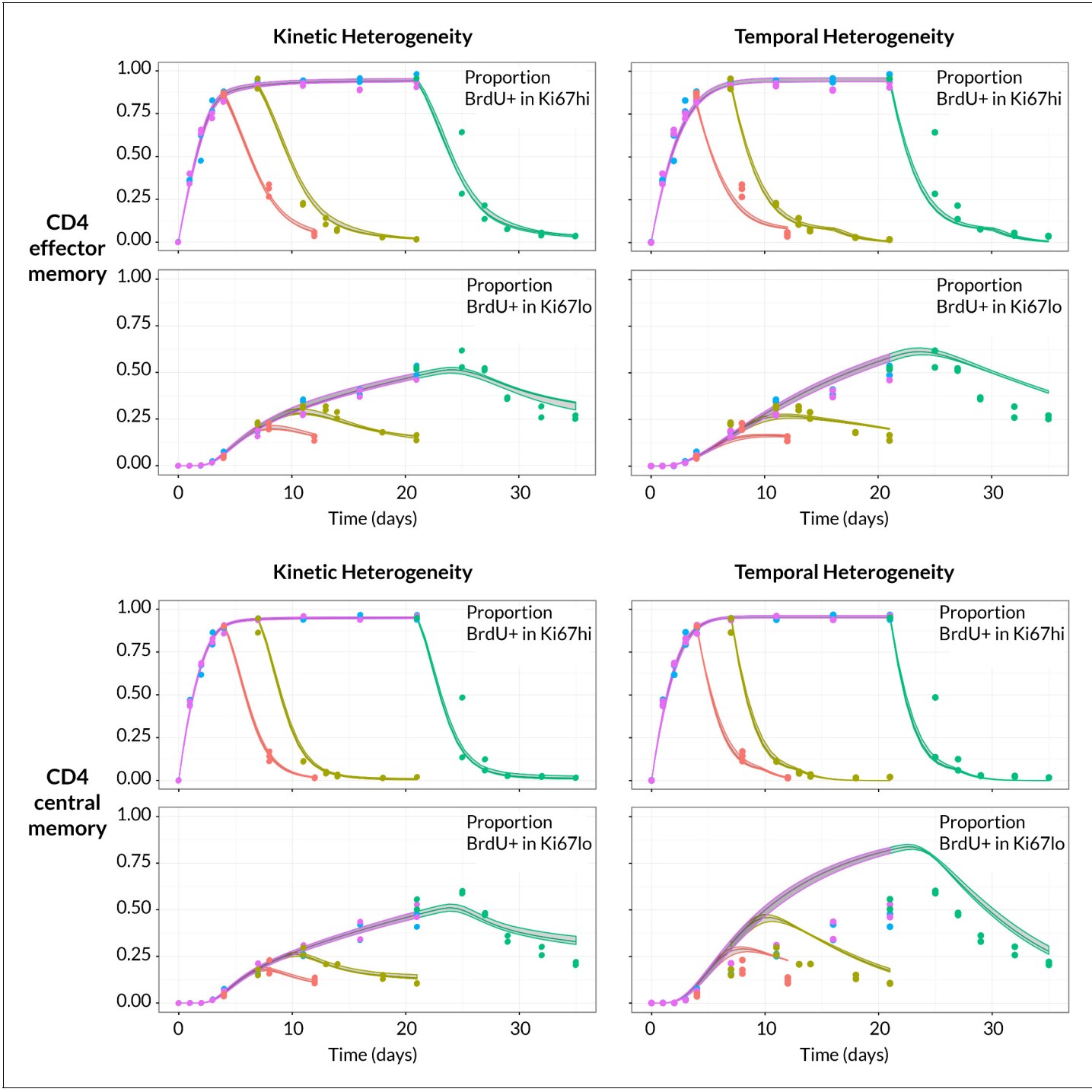

**Figure 4.** BrdU/Ki67 dynamics in memory CD4 T cell subsets are best described by a model of kinetically distinct subpopulations. Data and best fit predictions for two classes of model describing BrdU uptake and loss – kinetic heterogeneity (left panels) and temporal heterogeneity (right panels) – for CD4 $T_{EM}$ (upper panels) and CD4 $T_{CM}$ (lower panels). Fits were generated using the best-fit estimates of the influx into each population (for CD4 $T_{EM}$, 7.0% of the pool size per week at 14 weeks of age; for CD4 $T_{CM}$, 10.6% per week; these figures are 7 × the daily influx quoted in **Table 1**). Colours denote different BrdU feeding timecourses and shaded regions represent 95% confidence envelopes on the fits, calculated by resampling the parameters from their bootstrap distributions. The inability of the TH model to describe both the timecourses well stems from the tight coupling between the BrdU+Ki67$^{low}$ cells and their BrdU+Ki67$^{high}$ precursors, with little freedom to fit the timecourses of both simultaneously; whereas in the KH model, those two populations are enriched for the slow and fast subpopulations respectively, which are parameterised independently.

The following source data is available for figure 4:

*Figure 4 continued on next page*

*Figure 4 continued*

**Source data 1.** Timecourses of the BrdU$^+$ fractions within the Ki67$^{high}$ and Ki67$^{low}$ populations during the BrdU labelling/delabelling experiments.
**Source data 2.** Source code used to generate and fit models to BrdU/Ki67 timecourses.

## Sensitivity of predictions to the size of memory influx and choice of model

Given the notorious dependence of estimates of lymphocyte division and death rates on model assumptions (*De Boer and Perelson, 2013*), we explored the sensitivity of our estimates and predictions to the magnitude of the source. We performed fits to the BrdU/Ki67 timecourses for multiple

**Table 2.** Parameters describing homeostasis of murine CD4 memory subsets, using a two-population model of kinetic heterogeneity and the best estimates of the magnitudes of the influx into each subset from the naive pool. Pool-average lifetimes and interdivision times are defined to be the mean of the corresponding quantities for the fast and slow subpopulations weighted by their size estimates. The best-fitting models were those that assumed no difference in death rates of Ki67$^{high}$ and Ki67$^{low}$ cells (indicated by rows in which ratio of loss rates = 1).

| Parameter | Ratio of loss rates Ki67$^{high}$:Ki67$^{low}$ | CD4 effector memory Estimate | 95% CI | CD4 central memory Estimate | 95% CI |
|---|---|---|---|---|---|
| Pool-average lifetime (days) | 1 | 29 | (28, 30) | 21 | (19, 24) |
| | 10 | 41 | (39, 42) | 27 | (26, 35) |
| | 0.1 | 55 | (54, 56) | 44 | (39, 51) |
| Pool-average interdivision time (days) | 1 | 88 | (83, 158) | 86 | (47, 144) |
| | 10 | 63 | (53, 353) | 59 | (35, 74) |
| | 0.1 | 84 | (80, 108) | 51 | (38, 84) |
| Ki67 lifetime (days) | 1 | 3.28 | (3.14, 3.39) | 3.59 | (3.47, 3.70) |
| | 10 | 3.23 | (3.03, 3.30) | 3.74 | (3.57, 3.82) |
| | 0.1 | 3.28 | (3.18, 3.41) | 3.32 | (3.31, 3.52) |
| Efficiency of BrdU uptake (%) | 1 | 76 | (74, 79) | 77 | (76, 79) |
| | 10 | 75 | (73, 78) | 78 | (76, 79) |
| | 0.1 | 78 | (76, 81) | 78 | (78, 81) |
| Delay before source switches to BrdU$^-$ post-labelling (days) | 1 | 2.5 | (1.7, 3.0) | 0.085 | (0.002, 1.16) |
| | 10 | 2.6 | (1.9, 3.5) | 0.006 | (0.003, 1.43) |
| | 0.1 | 2.1 | (1.3, 2.3) | 0.14 | (0.003, 0.66) |
| Source contribution to peripheral production (fraction) | 1 | 0.12 | (0.12, 0.13) | 0.092 | (0.088, 0.096) |
| | 10 | 0.11 | (0.099, 0.11) | 0.085 | (0.08, 0.088) |
| | 0.1 | 0.15 | (0.15, 0.16) | 0.12 | (0.11, 0.12) |
| Fraction of source enteringslow subpopulation | 1 | 1 | (0.98, 1) | 0.69 | (0.38, 0.85) |
| | 10 | 1 | (0.95, 1) | 1 | (0.45, 1) |
| | 0.1 | 1 | (0.98, 1) | 0.46 | (0.19, 0.62) |
| Mean lifetime offast subpopulation (days) | 1 | 5.7 | (5.5, 6.8) | 3.3 | (3.1, 3.4) |
| | 10 | 17 | (15, 20) | 7.1 | (6.2, 7.5) |
| | 0.1 | 18 | (18, 21) | 6.4 | (6.4, 6.9) |
| Mean lifetime ofslow subpopulation (days) | 1 | 43 | (43, 48) | 38 | (34, 44) |
| | 10 | 61 | (60, 65) | 48 | (47, 65) |
| | 0.1 | 75 | (74, 77) | 76 | (68, 90) |
| Mean interdivision time offast subpopulation (days) | 1 | 5.7 | (5.5, 6.9) | 3.4 | (3.2, 3.5) |
| | 10 | 5.7 | (5.1, 6.7) | 3.3 | (3.0, 3.4) |
| | 0.1 | 6.8 | (6.8, 7.6) | 4.2 | (4.2, 4.4) |
| Mean interdivision time ofslow subpopulation (days) | 1 | 138 | (130, 275) | 167 | (89, 280) |
| | 10 | 113 | (90, 750) | 118 | (72, 151) |
| | 0.1 | 125 | (119, 169) | 92 | (69, 151) |
| Size of fast subpopulation(fraction of total) | 1 | 0.38 | (0.37, 0.44) | 0.49 | (0.47, 0.51) |
| | 10 | 0.46 | (0.43, 0.54) | 0.52 | (0.49, 0.53) |
| | 0.1 | 0.35 | (0.34, 0.38) | 0.47 | (0.45, 0.48) |

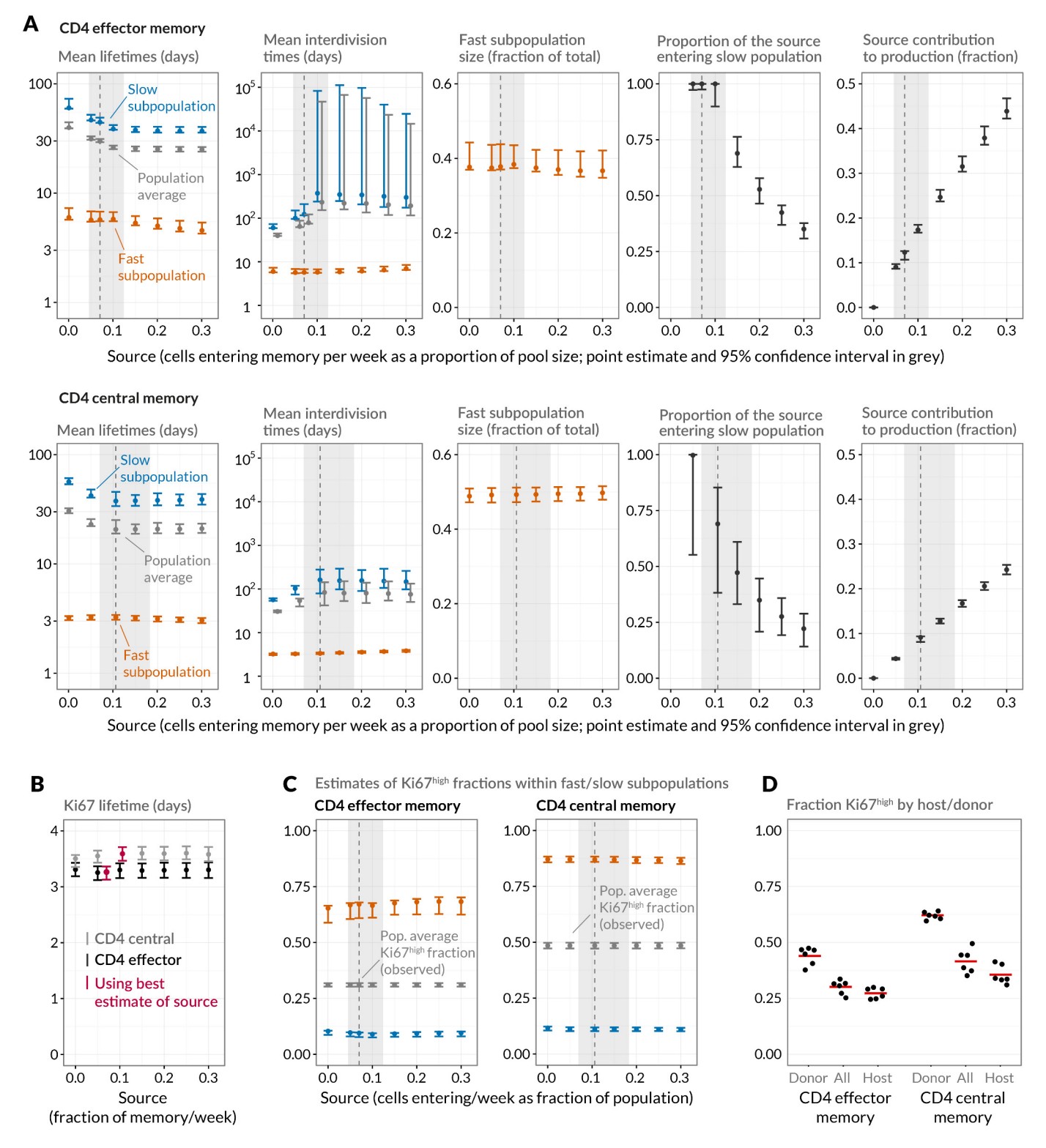

**Figure 5.** Quantifying CD4 $T_{EM}$ and CD4 $T_{CM}$ homeostasis assuming kinetic heterogeneity. (**A**) Key kinetic parameters for CD4 $T_{EM}$ and $T_{CM}$ estimated for different levels of memory influx. Grey points represent population average parameters; for interdivision times these are offset for clarity. Vertical dashed lines and shaded areas represent the best estimates of influx with 95% confidence intervals. These estimates are the weekly influxes as a fraction of the pool size (*i.e.*, 7 × the daily influxes quoted in *Table 1*; 0.07 of pool/week for CD4 $T_{EM}$, 0.11 for $T_{CM}$). (**B**) Estimated mean duration of Ki67 expression post-mitosis. (**C**) Estimated proportions of cells that are Ki67high within fast and slow subpopulations. The weighted averages of these

*Figure 5 continued on next page*

*Figure 5 continued*

proportions for each of CD4 $T_{EM}$ and CD4 $T_{CM}$ were constrained to be the observed level of expression (mean + s.e.m.) averaged over the course of the BrdU labelling experiments (*Appendix 1—figure 1*). (D) Stratifying Ki67$^{high}$ expression within CD4 $T_{EM}$ and CD4 $T_{CM}$ by host and donor, in six busulfan chimeras that were 8 weeks post-BMT and of comparable ages to the mice used in the BrdU labelling experiments, indicating that fast/slow cells cannot be exclusively identified as donor/host-derived.

The following source data and figure supplement are available for figure 5:

**Source data 1.** Ki67 expression in host and donor CD4 $T_{EM}$ and $T_{CM}$ cells in busulfan chimeras 8 weeks post-BMT (panel D).
**Figure supplement 1.** Comparing mean lifetimes and interdivision times obtained with the KH model when adding temporal heterogeneity.

values of the influx spanning values of zero to 30% of the pool size per week, which included the point estimates and their 95% confidence intervals (*Figure 5A*). Despite the two additional parameters required to describe the source (*i.e.* its partitioning between the fast and slow subpopulations, and the timing of the switch to unlabelled source after withdrawal of BrdU; see Appendix), including a source gave superior descriptions of both CD4 $T_{EM}$ and $T_{CM}$ labelling kinetics compared to models of self-renewing memory with no influx (ΔAIC = 9 and 20, respectively, at the best estimates of the source). The flows into memory impact measures of homeostatic dynamics significantly; if the contribution of the source is neglected, pool-averaged cell lifetimes may be overestimated by 25–50% and, more strikingly, interdivision times underestimated by a factor of 2–3 (*Figure 5A*).

At our best estimates of the influx into CD4 $T_{EM}$ and $T_{CM}$ from the naive pool, we infer that it predominantly feeds the slower subpopulations within each. Since we presume that memory is largely generated from naive cells through antigen-driven proliferation, this result was puzzling. A more restricted model in which the source was constrained to feed exclusively into the faster subpopulation had substantially lower statistical support (ΔAIC = 18 for CD4 $T_{EM}$, 9.9 for $T_{CM}$), but visually the fits were similar (*Appendix 1—figure 2*; parameter estimates in *Appendix 1—table 3*). Further, if CD4 $T_{EM}$ derive from $T_{CM}$ to any extent, we predict higher levels of influx (*Table 1*) and the proportion predicted to enter the slow population is then lower (*Figure 5A*, upper panels). We therefore remain cautious regarding the extents to which the constitutive influx feeds low and fast-dividing memory subsets. Irrespective, at all values of the source we explored, all variants of the KH model were far better descriptions of the kinetics than the TH model (ΔAIC > 90 for both CD4 $T_{EM}$ and $T_{CM}$).

Despite the richness of the BrdU/Ki67 timecourse, given the number of unknowns to be estimated it was not possible to fit a single model containing both forms of heterogeneity. However to look for a signature of temporal heterogeneity we explored variants of the KH model in which the loss rates of Ki67$^{high}$ cells were forced to be either a tenth or ten times that of the Ki67$^{low}$ cells in each subpopulation. For CD4 $T_{EM}$ neither extension improved on the basic KH model's description of the data (ΔAIC>4). For CD4 $T_{CM}$ we found almost equal support for a model in which Ki67$^{high}$ cells died 10 times faster than Ki67$^{low}$ cells and for the basic KH model in which death rates are independent of the level of Ki67 expression. This additional level of temporal heterogeneity increased the estimated mean lifetimes of both fast and slow CD4 $T_{CM}$ subsets but had little impact on estimates of interdivision times (*Table 2* and *Figure 5—figure supplement 1*). We conclude that our data do not provide evidence for substantial differences in the susceptibility to death of recently-divided and quiescent memory CD4 T cells.

We estimated the mean post-mitotic duration of Ki67 expression to be between 3.1 and 3.7 days, consistent with estimates elsewhere (*Pitcher et al., 2002*; *Younes et al., 2011*; *De Boer and Perelson, 2013*). This estimate was model-independent, insensitive to the magnitude of the influx into memory, and closely comparable for CD4 $T_{EM}$ and $T_{CM}$ (*Figure 5B*). The KH model predicted that the fast subpopulations express Ki67 at levels of approximately 65% (CD4 $T_{EM}$) and 85% ($T_{CM}$), while the slow populations in both are less than 10% Ki67$^{high}$ (*Figure 5C*). Ki67$^{high}$ CD4 memory cells are therefore predicted to be highly enriched for the fast dividing subset. Notably, the relative sizes of the fast and slow subsets were similar to the split of donor/host cells observed in memory in the busulfan chimeras (*Figure 2C*). It was then tempting to speculate that the slowly-dividing cells were the apparently resistant and stable populations of host-derived CD4 $T_{EM}$ and $T_{CM}$ cells in the

busulfan chimeras, while the more rapidly dividing cells represent the influx of donor cells into memory subsets. To test this, we measured Ki67 expression in busulfan chimeras 8 weeks after BMT, when hosts were a comparable age to those used in the BrdU feeding experiments. Although donor cells were indeed slightly enriched for Ki67$^{high}$ cells relative to host cells, both subpopulations exhibited substantial levels of Ki67 expression (*Figure 5D*) and did not map to the fast/slow populations inferred from the BrdU labelling analysis. These data therefore strongly suggest that both newly-recruited donor and more established host memory CD4 T cells are kinetically heterogeneous.

In summary, we find strong evidence for rapidly- and slowly-dividing populations within both effector and central CD4 memory T cells in uninfected adult mice. We find no strong evidence for recent division impacting susceptibility to cell death. Accounting for the constitutive flows of cells into both memory subsets significantly improves the description of BrdU labelling kinetics, and strongly impacts estimates of rates of memory T cell division and turnover.

## Discussion

To date, our understanding of how CD4 memory is structured and maintained has been limited by uncertainty in the interpretation of labelling data and lack of knowledge of the extent to which newly differentiated cells contribute to renewal. No single method has been able to successfully distinguish between and quantify these different processes. Here, we were able to both measure tonic influx into memory, and dissect memory compartment dynamics in detail by distinguishing turnover in quiescent and recently divided cells.

Temporal fate mapping in the busulfan chimeras revealed the surprisingly high rate of de novo generation of memory/effector cells in the CD4 memory compartments from naive cells, with at least 6–10% of cells replaced each week in 14 week old mice. Even at the lower bounds, the extent of this new memory generation from naive sources was surprising given that the hosts were in a clean, regulated environment and not deliberately infected. A recent study of feral mice and those in dirty environments revealed the expansion of CD8 T$_{EM}$ compartments resulting from the increased antigenic load (*Beura et al., 2016*). The authors concluded that expansion was driven by episodic exposure and not constitutive stimuli, as the activation and proliferative status of immune cells were similar to those in cleaner laboratory mice. However, our data strongly indicate the existence of tonic drivers of generation of new memory cells. An obvious mechanism is the continued recruitment of recent thymic emigrants into responses against commensal or environmental antigens. The fact that memory compartments remain remarkably stable in size in the face of this chronic stimulus suggests that these responding cells are regulated differently to those generated in an active infection, perhaps due to the absence of overt inflammatory stimulus. Whether inflammatory stimuli modulate these responses will be the subject of future study.

We also explored the differentiation pathways underlying the flow of cells from naive to different memory compartments over timescales of weeks to months. Previous studies suggest that regulation of CD62L expression by activated CD4 T cells is both heterogenous and slow, compared with CD8 cells (*Bjorkdahl et al., 2003*; *Chao et al., 1997*). Loss of CD62L expression is largely irreversible in CD4 T$_{EM}$ (*Kassiotis and Stockinger, 2004*; *Bingaman et al., 2005*), suggesting that the CD62L-expressing CD4 T$_{CM}$ derive directly from activation of naive T cells and not from T$_{EM}$. Consistent with this we clearly observed more rapid and slightly greater replacement of the CD4 T$_{CM}$ than CD4 T$_{EM}$ compartment in busulfan chimeras. For CD4 T$_{EM}$ the situation is less clear, but CD8 T$_{EM}$ may be generated both directly from activation of naive T cells or by subsequent differentiation of CD8 T$_{CM}$ (*Restifo and Gattinoni, 2013*). Due to the risk of overfitting it was not possible to quantify the contributions of each of these pathway to CD4 T$_{EM}$ at steady state, and so we considered only the extreme alternatives in which CD4 T$_{EM}$ are sourced entirely from naive or entirely from CD4 T$_{CM}$. The T$_{CM}$ → T$_{EM}$ model was statistically inferior but gave visually similar fits (*Figure 2C*), and predicted much higher rates of CD4 T$_{EM}$ replacement (~23%/week, compared to ~6% for a naive source; *Table 1* ). As it seems likely that the CD4 T$_{EM}$ population is fed by both naive and T$_{CM}$ cells to some extents, we conclude that our estimate of 6% is a lower bound and it is possible that nearly a quarter of CD4 T$_{EM}$ are replaced each week under healthy conditions at 14 weeks of age.

Our estimated average lifetimes of lymph-node-derived memory CD4 T cells (21d for T$_{CM}$ and 29d for T$_{EM}$) are slightly higher than those made previously. Other studies of total CD44$^{hi}$ CD4 T cells using BrdU or deuterated water labelling found kinetics consistent with mean lifetimes of 14-

22d (*De Boer and Perelson, 2013*; *Westera et al., 2013*). One of these studies found that a simple model of two self-renewing, stable populations described the labelling kinetics better than the simplest single-compartment model with no temporal heterogeneity, and that the fast and slow CD4 memory populations were comparable in size (*Westera et al., 2013*). Both studies assumed that memory is self-renewing and constant in size, so mean lifetimes are necessarily equal to average interdivision times. If we make a similar assumption and neglect the memory sources, our estimates of lifetimes increase (40d and 30d for CD4 $T_{EM}$ and $T_{CM}$ respectively). Thus, studies assuming memory is a self-renewing compartment will tend to overestimate lifetimes and underestimate interdivision times. This issue again highlights the sensitivity of measures of population dynamics to the biology encoded in the model. Our analysis suggests that the rate of recruitment into memory from the naive pool varies with age, and given the relative stability of memory population sizes it is therefore likely that memory turnover is also not constant over the life course. It is possible that the discrepancies between our and other estimates may derive from differences in host age or in commensal colonisation arising in the different housing facilities, both of which may impact the rate of tonic recruitment into memory, the relative sizes of fast and slowly dividing subpopulations, and hence estimates of cell lifetimes and division rates.

Our experimental analyses revealed heterogeneous behaviour amongst memory CD4 T cells at multiple levels. Heterogeneity within the CD4 memory compartment as a whole has been recognised for some time, with a distinction drawn between slow-dividing cells driven by cytokines (*Seddon et al., 2003*; *Purton et al., 2007*) and in which memory to defined antigens is thought to reside, and fast-dividing cells (*Tough and Sprent, 1994*; *Robertson et al., 2006*; *Purton et al., 2007*; *Surh and Sprent, 2008*; *Younes et al., 2011*) whose proliferation is dependent on TCR signalling (*Min et al., 2003*; *Seddon et al., 2003*; *Leignadier et al., 2008*; *Younes et al., 2011*). We infer that Ki67 expression levels in the slower-dividing populations are approximately 9–11% (*Figure 5C*), consistent with direct observations of antigen-specific CD4 memory (*Lenz et al., 2004*; *Purton et al., 2007*; *Pepper et al., 2010*; *Younes et al., 2011*). However, the relation of these kinetically distinct populations to the canonical CD4 $T_{EM}$ and $T_{CM}$ subsets delineated by CD62L expression has been unclear. Here, we find evidence that CD4 $T_{EM}$ and $T_{CM}$ subsets comprise both fast and slow subpopulations, suggesting that they are indeed similar in their homeostatic dynamics and structure. The lineage relationships between the respective fast and slow subpopulations of CD4 $T_{EM}$ and $T_{CM}$ therefore need further investigation.

In the absence of infection, it is intuitive that the stimulus driving the continuous recruitment into memory derives from environmental antigens in food and commensal organisms. It also seems intuitive that this stimulus should continue to drive fast-dividing memory subpopulations throughout life. Certainly, we find evidence of these within both new (donor-derived) and more established (host-derived) memory populations (*Figure 5D*). However if environmental antigens are the stimulus for tonic recruitment, then it is surprising that newly generated memory cells are not exclusively fast-dividing. These observations could be explained if exposure to environmental antigens is subject to natural fluctuations in load, resulting in episodic but frequent stimuli to divide rather than continuous rounds of division. Indeed bursts of TCR-driven proliferation may be involved in the maintenance of CD4 memory to persistent phagosomal infections (*Nelson et al., 2013*). Such a view would be consistent with the estimates of interdivision times for the fast subpopulations, which are still much longer than the interdivision times of several hours that result from cognate antigen challenge. Episodic fast divisions within both CD4 $T_{EM}$ and $T_{CM}$ could also account for the arguably counter-intuitive model prediction that the source predominantly feeds the slowly-dividing subpopulations. We also note that the fits yielded by the model of discrete fast and slow populations are good, but not perfect, and it seems likely that there is a richer kinetic substructure (*Ganusov et al., 2010*). Testing this hypothesis that the composition of fast and slow populations is dynamic, establishing how the influx into memory is routed to these subpopulations, and identifying the lineage relationships between CD4 $T_{EM}$ and $T_{CM}$ at steady state, will require new approaches. What is clear, however, is that whether subdivided by surface phenotype or age structure, kinetically distinct subpopulations are consistently demonstrable within memory CD4 T cells.

Our analysis also provides new insight into the interpretation of Ki67 expression, which is commonly used as a proxy for levels of cell proliferation. There has been a growing awareness that while Ki67 is induced at onset of cell cycle, expression persists following completion of mitosis (*Pitcher et al., 2002*; *Younes et al., 2011*; *Hogan et al., 2013*; *De Boer and Perelson, 2013*). Here

we explicitly model its expression and estimate that cells take approximately 3.5 days to become Ki67$^{low}$. This figure depends in part on the flow cytometry gating strategy and so is a functional rather than a biochemical measure. However the modelling indicated that the residence time in Ki67$^{high}$ has a low coefficient of variation and so we infer that the post-mitotic loss of Ki67 is essentially deterministic, with very little cell-cell variation. Since cell division only takes between 2–8 hr (*Bruno and Darzynkiewicz, 1992*; *Hogan et al., 2013*), Ki67 is therefore chiefly a post-mitotic marker. Its extended expression makes it a sensitive measure for detecting cell division occurring at low absolute frequencies. Knowledge of its lifetime is also useful for isolating cell populations. The fast CD4 $T_{CM}$ and $T_{EM}$ subpopulations divide approximately every 3 and 6 days respectively, while their slower counterparts divide only every 140 days or more. Therefore a substantial fraction of the fast subsets (greater than half for CD4 $T_{CM}$) will begin to divide again before losing Ki67 expression. Ki67-bright CD4 memory cells are therefore highly enriched for the fast dividing subsets, and the BrdU$^+$ Ki67$^{low}$ subset is increasingly rich in slowly-dividing cells. These properties can be used as basis for further functional characterisation of these subpopulations.

Taken together, our data reveal complexity in the regulation of memory compartments, in which the substantial and tonic de novo generation of memory cells braids into highly dynamic and heterogeneous subpopulations which themselves exhibit an unexpectedly diverse age structure. Despite this complexity in cell dynamics, the compartment sizes are remarkably stable throughout life, indicating tight homeostatic control. Key questions for the future are whether tonic influxes contribute to the erosion of antigen-specific CD4 T cell memory over time (*Homann et al., 2001*), whether the tonic recruitment and turnover of memory cells are modulated during the course of an inflammatory, infectious episode, and whether this backdrop of memory cell activity in any way influences T cell activation and development that occurs during such challenges.

# Methods and materials

## Experimental protocols

### Mice

WT CD45.1 and CD45.2 mice were bred and maintained in conventional pathogen-free colonies at either the National Institute for Medical Research (London, UK) or at the Royal Free Campus of University College London. All experiments were performed in accordance with UK Home Office regulations, project license number PPL70-8310.

### Busulfan chimeras

Chimeric mice were generated as described previously in Bio-protocol (*Hogan et al., 2017a*). Briefly, WT CD45.1 mice aged 8 weeks were treated with 20 mg/kg busulfan (Busilvex, Pierre Fabre) to deplete HSC, and reconstituted with T-cell depleted bone marrow cells from congenic donor WT CD45.2 mice. Chimeras were sacrificed at 6–52 weeks after bone marrow transplantation, and cells from the thymus, spleen and lymph nodes were analysed by flow cytometry.

### BrdU timecourses

BrdU (Sigma) was administered to WT mice by an initial intraperitoneal injection of 0.8 mg BrdU, followed by maintenance of 0.8 mg/mL BrdU in drinking water for the indicated time periods up to 21 days. BrdU in drinking water was refreshed every 2–3 days. Starting times for BrdU treatment were staggered so that all mice in a timecourse experiment were sacrificed on the same day for analysis of lymph node cells by flow cytometry. Ages at sacrifice were in the range 14–16 weeks. The protocol described in more detail at Bio-protocol (*Hogan et al., 2017b*).

### Flow cytometry

Cells were stained with the following monoclonal antibodies and cell dyes: CD45.1 FITC, CD45.2 AlexaFluor 700, TCR-beta APC, CD4 PerCP-eFluor710, CD25 PE, CD44 APC-eFluor780, CD25 eFluor450, CD62L eFluor450 (all eBioscience), TCR$\beta$ PerCP-Cy5.5, CD5 BV510, CD4 BV650, CD44 BV785 (all BioLegend), CD62L BUV737 (BD Biosciences), LIVE/DEAD nearIR and LIVE/DEAD Blue viability dyes (Invitrogen). BrdU and Ki67 co-staining was performed using the FITC BrdU Flow Kit (BD

Biosciences) according to the manufacturer's instructions, along with anti-Ki67 eFluor660 (eBioscience). Cells were acquired on a BD LSR-II or BD LSR-Fortessa flow cytometer and analysed using Flowjo software (Treestar). Subset gates were as follows: CD4 naive: live TCR$\beta$+ CD5+ CD4+ CD25-CD44- CD62L+. CD4 $T_{EM}$: live TCR$\beta$+ CD5+ CD4+ CD25- CD44+ CD62L-. CD4 $T_{CM}$: live TCR$\beta$+ CD5+ CD4+ CD25- CD44+ CD62L+.

## Modelling the fluxes between naive, central memory and effector memory subsets

We used a simple framework to describe the kinetics of constitutive renewal of the effector and central memory CD4 compartments. Assume that cells flow into a memory subset $M(t)$ at *per capita* rate $\gamma$ from a precursor population $S(t)$, and that in the absence of this source, memory is lost to death and/or differentiation at net rate $\lambda$. We place no constraints on the growth or decay of memory so $\lambda$ may be positive or negative. Then if host and donor cells follow identical kinetics,

$$\frac{dM_{\text{host}}}{dt} = \gamma S_{\text{host}}(t) - \lambda M_{\text{host}}(t) \tag{1}$$

$$\frac{dM_{\text{donor}}}{dt} = \gamma S_{\text{donor}}(t) - \lambda M_{\text{donor}}(t) \tag{2}$$

The rate $\gamma$ is the product of the *per capita* rate of egress of cells from the precursor population $S$ and the net effect of any expansion and/or contraction that takes place during the transition into memory. We define the memory chimerism to be the fraction of cells that are donor-derived,

$$\chi_{\text{M}} = \frac{M_{\text{donor}}}{M_{\text{donor}} + M_{\text{host}}}$$

which differs among age-matched animals due to variation in the degree of HSC depletion with busulfan treatment. We normalise the memory chimerism to that in the thymic precursor population DP1, $\chi_{\text{DP1}}$, which is stable by approximately 6 weeks post-BMT. *Equations 1 and 2* can then be recast in forms that do not depend on the degree of HSC depletion and so are applicable across mice in the experimental cohort:

$$\frac{dM}{dt} = \gamma S(t) - \lambda M(t) \tag{3}$$

$$\frac{d\rho_{\text{M}}}{dt} = \frac{\gamma S(t)}{M(t)} (\rho_S(t) - \rho_{\text{M}}(t)) \tag{4}$$

where $M = M_{\text{host}} + M_{\text{donor}}$, and $\rho_M = \chi_{\text{M}}/\chi_{\text{DP1}}$ and $\rho_S = \chi_S/\chi_{\text{DP1}}$ are the normalised donor chimerism in the memory and source populations respectively.

To account for the apparent capping of chimerism in the memory subsets, this model can be extended to allow the per-cell rate of recruitment into memory to vary with the age of the animal, $\gamma(t)$, and/or a population of host-derived memory cells, $M_{\text{inc}}$, that resists displacement by newer cells. Combining these extensions yields

$$\frac{dM}{dt} = \gamma(t)S(t) - \lambda (M(t) - M_{\text{inc}}) \tag{5}$$

$$\frac{d\rho_{\text{M}}}{dt} = \frac{\gamma(t)S(t)}{M(t)} (\rho_S(t) - \rho_{\text{M}}(t)) - \frac{\lambda M_{\text{inc}}}{M(t)} \rho_{\text{M}}(t) \tag{6}$$

where now $M = M_{\text{host}} + M_{\text{donor}} + M_{\text{inc}}$. Note that in the text we work with two sub-models – one in which there is a resistant population $M_{\text{inc}}$ but the *per capita* rate of recruitment from the source $\gamma = \gamma_0$ is a constant; and another in which all memory is displaceable ($M_{\text{inc}} = 0$) but that the *per capita* rate of recruitment from the source wanes with age, $\gamma(t) = \gamma_0 \exp(-\phi t)$. Below we show the predictions of the most general model that combines both elements.

We can make a conservative estimate of the flows into memory by assuming that both the effector and central memory pools are fed directly from the naive pool. For each, we fitted the model of the dynamics of total cell numbers and the normalised chimerism within the naive CD4 T cell pool (*Equations 5 and 6*) to the observations, using empirical functions describing changes in the size

and chimerism of the source population with time. The following functional forms described both putative source populations (naive and CD4 $T_{CM}$) well;

Source population size

$$S(t) = S_0 e^{-Rt} \tag{7}$$

Source chimerism

$$\rho_S(t) = \frac{\rho_{max}}{1 + e^{-rt}(\rho_{max} - \rho_0)/\rho_0} \tag{8}$$

Estimates of the parameters governing these functions are given in *Appendix 1—table 1*.

To fit the resistant memory and declining recruitment models, we maximised the product of the log likelihoods of the two timecourses, as described in the Supporting Information of *Hogan et al. (2015)*, using a trust region method implemented in Python (*Figure 2—source data 2*). We estimated the net rate of loss/growth of memory, $\lambda$ and the initial rate of recruitment from the source, $\gamma_0$; and either (i) $M_{inc}$ for the model of resistant memory with constant *per capita* rate of recruitment from the source; or (ii) $\phi$ for the model in which the *per capita* rate of recruitment from the source itself falls exponentially at rate $\phi$. Confidence intervals on these parameters were generated by simultaneously bootstrapping residuals 3000 times and resampling from the bootstrap estimates of the parameters governing the sources, $S(t)$ and $\rho_S(t)$.

One quantity of interest is $\gamma(t)S(t)/M(t)$, which is the number of new cells entering memory per unit time as a fraction of the memory pool size. Another is the fraction of memory cells replaced through immigration. Because we assumed that recently recruited memory cells are as susceptible to loss as older displaceable memory cells, the number of memory cells expected to be replaced through immigration during a time $(0, t)$ is slightly less than the total influx in the same period. Consider a memory population $M(0)$ that comprises a displaceable population $X(0)$ and an incumbent self-renewing population $M_{inc}$, $M(0) = X(0) + M_{inc}$. We defined the fractional replacement to be the proportion of the memory pool comprised of immigrants after a time $t$;

$$f_{replace}(t) = \frac{Y(t)}{X(t) + Y(t) + M_{inc}} \tag{9}$$

where $X(t) = X(0)e^{-\lambda t}$ is the number of displaceable cells present at $t = 0$ which survived to time $t$; and $Y(t)$ is the number of cells that entered memory during $(0, t)$ and survived to $t$, which is the solution to $dY/dt = \gamma(t)S(t) - \lambda Y(t)$ given $Y(0) = 0$. If we assume $\gamma(t) = \gamma_0 \exp(-\phi t)$ and the empirical form for the source $S(t) = S_0 \exp(-Rt)$, this yields

$$f_{replace}(t) = \frac{\gamma_0 S_0 (e^{\psi t} - 1)}{e^{\psi t}(\psi(M(0) - M_{inc}) + \gamma_0 S_0) + \psi M_{inc} e^{(\psi + \lambda)t} - \gamma_0 S_0}, \tag{10}$$

where we define $\psi = R + \phi - \lambda$. In the text we quote both the daily influx $\gamma S/M$ and the expected weekly fractional replacement, $f_{replace}(t = 7 \text{ days})$, both at 14 weeks of age, which is 6 weeks post-BMT in these animals (*Table 1*). We also quote replacement as a fraction of the displaceable sub-population only:

$$f_{replace}^{displaceable}(t) = \frac{Y(t)}{X(t) + Y(t)} = \frac{\gamma_0 S_0 (e^{\psi t} - 1)}{e^{\psi t}(\psi(M_0 - M_{inc}) + \gamma_0 S_0) - \gamma_0 S_0}. \tag{11}$$

*Figure 2D* shows how the weekly fractional replacement of both total and displaceable memory is predicted to change with age.

The estimates of daily influx are robust to the details of the model of the capping of memory chimerism. This is because information regarding this parameter is largely contained in the gradient of the donor chimerism in memory $d\rho_M/dt$ early in reconstitution, which is well-defined in the data (*Figure 2C*, lower panels; animals of age *c.* 100 days). When the chimerism $\rho_M$ and/or the rate of change of memory cell numbers in the absence of influx ($\lambda$) are low, then in either model

$$\frac{\gamma(t)S(t)}{M(t)} \simeq \frac{d\rho_M/dt}{\rho_S(t) - \rho_M(t)} = \frac{d\chi_M/dt}{\chi_S - \chi_M}. \tag{12}$$

This expression yields estimates of influx of 2.5% of the pool size per day for CD4 $T_{CM}$ and 1.2% per day for CD4 $T_{EM}$, assuming both are sourced by naive cells. Both of these estimates lie within the 95% confidence intervals calculated using the best-fitting models (*Table 1*). *Equation 12* also explains why the predicted rates of CD4 $T_{EM}$ replacement are much higher if one assumes that the source is $T_{CM}$ rather than naive; this rate scales inversely with the difference in chimerism between the source and target populations, which is lower for a $T_{CM} \rightarrow T_{EM}$ pathway than for naive $\rightarrow T_{EM}$.

## Modelling BrdU/Ki67 dynamics

The model is illustrated in *Figure 3C*, indicating flows between the $BrdU^{-/+} \times Ki67^{low/high}$ populations following first order kinetics with fixed rate constants. To describe the data we made the following extensions to this basic structure:

1. *Figure 3C* represents a cell population exhibiting temporal heterogeneity; all cells share a common *per capita* rate of entry into division ($\alpha$) but quiescent ($Ki67^{low}$) and dividing or recently divided cells ($Ki67^{high}$) have potentially different rates of death ($\delta^-$ and $\delta^+$ respectively). Kinetic heterogeneity can be represented by combining two or more instances of this model, with each subpopulation present in unknown proportions and with potentially different rate constants and/or source terms. To minimise the number of parameters, and motivated by our analysis of the pathways of the fluxes into memory, we assumed that there are no significant flows between these subpopulations.
2. $Ki67^{high}$ cells are shown transitioning to $Ki67^{low}$ with first-order kinetics at rate $\beta$. Ki67 expression is continuous, however (*Figure 3A*) and we model its loss post-mitosis as a multi-step process through intermediate expression levels, giving gamma-distributed residence times in the $Ki67^{high}$ compartments with mean $1/\beta$. The variance in the residence time is unknown and we explored it by modelling $\hat{k}$ intermediate steps between the highest and lowest levels of Ki67 expression (*i.e.* from immediately post-mitosis to quiescence), with a transition rate of $\hat{k}\beta$ between each of them. Both $\hat{k}$ and $\beta$ are parameters to be estimated. This distribution of residence times reflects a combination of the kinetics of Ki67 loss post-mitosis and the gating strategy used to distinguish $Ki67^{low}$ and $Ki67^{high}$ cells.
3. For simplicity *Figure 3C* indicates that during administration all dividing cells are labelled with BrdU, but its efficiency of uptake is a free parameter ($\epsilon$, with $0 < \epsilon \leq 1$).
4. Following withdrawal of BrdU, we assumed that label is diluted through division. The number of divisions required for a $BrdU^+$ cell to return to unlabelled status ($BrdU^-$) is a free parameter in the model, $\hat{b}$. Thus we modelled multiple sub-compartments within the $BrdU^+$ population. If BrdU uptake is not 100% efficient ($\epsilon < 1$), BrdU dilution can also occur during labelling, through division of previously labelled cells without further uptake.
5. We assumed that any source-derived cells are $Ki67^{high}$ and $BrdU^+$ during labelling, and that the source switches to $BrdU^-$ a time $\tau$ after labelling stops.

With these extensions, the model parameterises the following processes:

$\alpha$ = rate of entry into division resulting in expression of Ki67,

$\delta^-$ = rate of loss (death or differentiation) for $Ki67^{low}$ cells,

$\delta^+$ = rate of loss (death or differentiation) for $Ki67^{high}$ cells,

$\epsilon$ = probability of incorporating BrdU per division during label administration,

$S^\pm$ = rate of entry of cells into the $K^+B^\pm$ compartments from the source,

$\hat{k}$ = number of $Ki67^{high}$ intermediate compartments,

$1/\beta$ = mean duration of Ki67 expression (i.e., mean total residence time in $Ki67^{high}$ compartments),

$\hat{b}$ = number of divisions required for a $BrdU^+$ cell to become $BrdU^-$ in the absence of label,

$\tau$ = time for source to become $BrdU^-$ following withdrawal of BrdU.

For the basic KH model we set $\delta^+ = \delta^- = \delta$ for each subpopulation. We also examined variants of KH in which the death rate of $Ki67^{high}$ cells in each subpopulation was forced be either 1/10 or 10 times the death rate of of $Ki67^{low}$ cells.

Detailed descriptions of the representation of this model as ordinary differential equations and the procedure for parameter estimation are given in Appendix 1.

## Acknowledgments

The analyses were supported in part through the computational resources and staff expertise provided by Scientific Computing at the Icahn School of Medicine at Mount Sinai.

## Additional information

### Funding

| Funder | Grant reference number | Author |
|---|---|---|
| National Institutes of Health | R01 AI093870 | Andrew J Yates |
| Arthritis Research UK | | Andrew J Yates |
| Medical Research Council | MC-PC-13055 | Thea Hogan<br>Benedict Seddon |
| National Science Foundation | 1548123 | Graeme Gossel |

The funders had no role in study design, data collection and interpretation, or the decision to submit the work for publication.

### Author contributions

GG, Software, Formal analysis, Investigation, Visualization, Methodology, Writing—original draft; TH, Data curation, Formal analysis, Investigation, Methodology; DC, Software, Formal analysis, Investigation, Visualization, Methodology, Writing—review and editing; BS, Conceptualization, Supervision, Funding acquisition, Investigation, Visualization, Methodology, Writing—original draft, Writing—review and editing; AJY, Conceptualization, Resources, Formal analysis, Supervision, Funding acquisition, Investigation, Visualization, Methodology, Writing—original draft, Writing—review and editing

### Author ORCIDs

Andrew J Yates, http://orcid.org/0000-0003-4606-4483

### Ethics

Animal experimentation: All experiments were performed in accordance with UK Home Office regulations, project license number PPL70-8310.

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

# Appendix 1

**Appendix 1—table 1.** Parameter estimates for the functions used to empirically describe the timecourses of the size and chimerism of the putative source populations feeding memory CD4 T cell subsets.

| | Exponential model of source numbers | |
| --- | --- | --- |
| | $S(t) = S_0 \exp(-Rt)$ | |
| | Naive source | $T_{CM}$ source |
| $S_0 R$ | 5.2 (4.4, 6.2) $\times 10^7$ | 1.1 (0.81, 1.4) $\times 10^6$ |
| $R$ | 5.3 (4.5, 6.0) $\times 10^{-3}$ | 2.3 (0.80, 3.6) $\times 10^3$ |

| | Logistic model of normalised source chimerism | |
| --- | --- | --- |
| | $\rho_S(t) = \rho_{max}/(1 + \exp(-rt)(\rho_{max} - \rho_0)/\rho_0)$ | |
| | Naive source | $T_{CM}$ source |
| $\rho_{max}$ | 0.85 (0.80, 0.88) | 0.48 (0.41, 0.56) |
| $\rho_0$ | 0.09 (0.012, 0.18) | 0.013 (0.0032, 0.034) |
| $r$ | 0.038 (0.027, 0.080) | 0.045 (0.030, 0.073) |

**Appendix 1—table 2.** Parameter estimates for the models describing the feeding of memory CD4 T cell subsets from a source. Parameters common to both models are defined as follows: $M_0$, total (host + donor) initial number of memory cells; $\rho_0$, the chimerism in the memory compartment at age 14 weeks, normalised to that at the DP1 stage of thymic development; $\gamma_0$, the *per capita* conversion rate of source cells into memory cells at age 14 weeks; $\lambda$, the *per capita* loss rate of displaceable memory cells through death and differentiation. Parameters specific to the resistant memory model: $M_{inc}$, the number of resistant, host-derived memory cells, assumed to be constant over time. For the declining recruitment model: $\phi$, the rate of decay of the *per capita* rate of conversion of source cells into memory, with age (yielding $\gamma(t) = \gamma_0 \exp(-\phi t)$). AIC differences are quoted relative to the best-fitting model for each population.

| | Resistant memory model | | |
| --- | --- | --- | --- |
| | CD4 $T_{EM}$ | | CD4 $T_{CM}$ |
| | Naive source | $T_{CM}$ source | Naive source |
| $M_0$ | 5.5 (4.6, 6.8) $\times 10^6$ | 4.3 (3.2, 5.8) $\times 10^6$ | 0.93 (0.68, 1.3) $\times 10^6$ |
| $\rho_0$ | 0.026 (0.019, 0.035) | 0.027 (0.019, 0.039) | 0.074 (0.051, 0.11) |
| $\gamma_0$ | 1.7 (1.1, 2.5) $\times 10^{-3}$ | 0.20 (0.13, 0.40) | 4.5 (2.8, 8.7) $\times 10^{-4}$ |
| $\lambda$ | 1.6 (−1.9, 5.4) $\times 10^{-3}$ | 0.012 (0.0060, 0.034) | 0.019 (0.011, 0.038) |
| $M_{inc}$ | 5.3 (0.0074, 6.1) $\times 10^6$ | 6.2 (0.00023, 24.0) $\times 10^5$ | 1.5 (0.75, 2.2) $\times 10^5$ |
| **ΔAIC** | 0.16 | 11.0 | 0.0 |

| | Declining recruitment model | | |
| --- | --- | --- | --- |
| | CD4 $T_{EM}$ | | CD4 $T_{CM}$ |
| | Naive source | $T_{CM}$ source | Naive source |
| $M_0$ | 5.6 (4.5, 6.8) $\times 10^6$ | 4.4 (3.3, 6.0) $\times 10^6$ | 6.9 (5.0, 9.3) $\times 10^5$ |
| $\rho_0$ | 0.026 (0.018, 0.035) | 0.028 (0.018, 0.040) | 0.073 (0.053, 0.11) |
| $\gamma_0$ | 2.2 (0.94, 5.8) $\times 10^{-3}$ | 0.17 (0.096, 0.37) | 1.3 (0.47, 4.0) $\times 10^{-3}$ |
| $\lambda$ | 0.22 (−1.3, 1.7) $\times 10^{-3}$ | 0.011 (0.0042, 0.025) | 7.1 (5.1, 9.6) $\times 10^{-3}$ |

*Appendix 1—table 2 continued on next page*

*Appendix 1—table 2 continued*

| | **Resistant memory model** | | |
| --- | --- | --- | --- |
| $\phi$ | **2.2** (−2.5, 8.3) ×$10^{-3}$ | **−0.85** (−4.2, 5.0) × $10^{-3}$ | **9.7** (3.8, 17.0) × $10^{-3}$ |
| $\Delta$**AIC** | 0.0 | 11.0 | 8.3 |

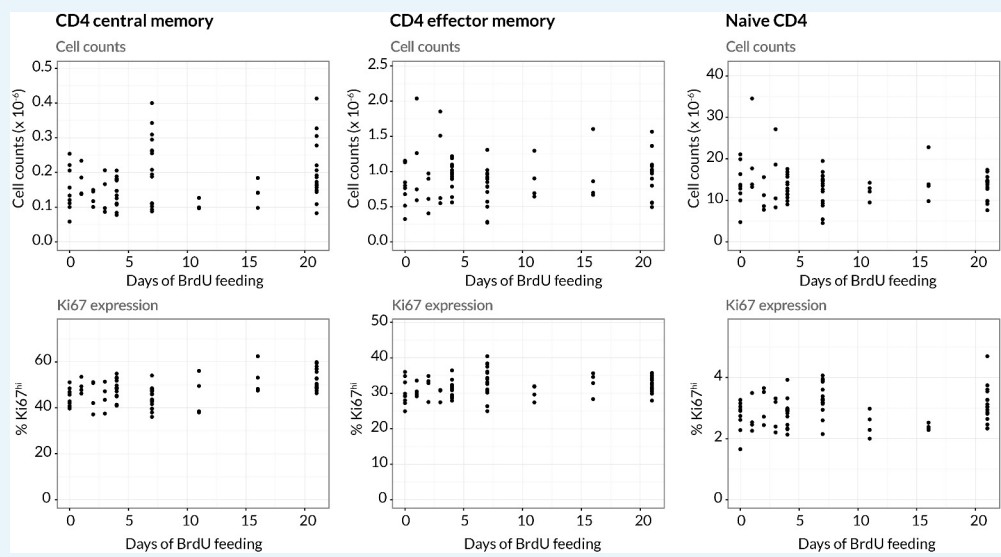

**Appendix 1—figure 1.** Stability of cell numbers and Ki67 expression during BrdU labelling. Cells for BrdU/Ki67 analysis were recovered from lymph nodes. We show the corresponding data for naive CD4 T cells to confirm that BrdU feeding has no significant impact on the (putative) source population.

**Appendix 1—figure 1—source data 1.** Data showing stability of both numbers and Ki67 expression of effector memory, central memory and naive CD4 T cells recovered from lymph nodes during BrdU labelling.

# Detailed description of modelling of BrdU/Ki67 timecourses

We define $B_0$ and $K_0$ to be the numbers of BrdU$^-$ and Ki67$^{low}$ cells respectively; $B_{\hat{b}}$ and $K_{\hat{k}}$ to be the numbers of cells with the highest levels of expression of BrdU and Ki67 respectively; and $B_i$ ($0 < i < \hat{b}$) and $K_j$ ($0 < j < \hat{k}$) to be populations expressing intermediate levels of BrdU or Ki67, but still classed as BrdU$^+$ or Ki67$^{high}$ in terms of our flow cytometry gating strategy. A cell that has successfully incorporated BrdU must therefore undergo $\hat{b}$ further divisions without further label uptake to become BrdU$^-$. Similarly, $\hat{k}$ is the number of sub-states through which post-mitotic Ki67$^{high}$ cells must transition before becoming Ki67$^{low}$. Each transition occurs with first-order kinetics at rate $\hat{k}\beta$, such that the total time spent in the Ki67$^{high}$ states $\{\hat{k}, \dots, 1\}$ is gamma-distributed with mean $1/\beta$. When fitting the models we explored multiple values of $\hat{b}$ and $\hat{k}$. For all models of heterogeneity and magnitudes of the source, $\hat{b} = 2$ and $\hat{k} > 12$ yielded the lowest AIC values. The former indicates that two divisions without label uptake are sufficient to dilute into the BrdU$^-$ gate. The latter indicates that there is a narrow distribution of times spent in Ki67$^{high}$ post-mitosis. We then

use $B_iK_j$ to denote the number of cells simultaneously at the $i^{\text{th}}$ level of BrdU expression and the $j^{\text{th}}$ level of Ki67 expression, leading to the following definitions:

$$
\begin{aligned}
\mathrm{BrdU}^+\mathrm{Ki67}^{\text{high}} &= \sum_{i=1}^{\hat{b}}\sum_{j=1}^{\hat{k}} B_iK_j \\
\mathrm{BrdU}^+\mathrm{Ki67}^{\text{low}} &= \sum_{i=1}^{\hat{b}} B_iK_0 \\
\mathrm{BrdU}^-\mathrm{Ki67}^{\text{high}} &= \sum_{j=1}^{\hat{b}} B_0K_j \\
\mathrm{BrdU}^-\mathrm{Ki67}^{\text{low}} &= B_0K_0.
\end{aligned}
\tag{13}
$$

Dropping the time-dependence of the cell populations for brevity, the set of coupled ODEs describing a single population of cells entering division at rate $\alpha$, and with Ki67$^{\text{low}}$ and Ki67$^{\text{high}}$ cells being lost through death or differentiation at rates $\delta^-$ and $\delta^+$ respectively, is

$$
\begin{aligned}
\tfrac{d}{dt}B_{\hat{b}}K_{\hat{k}} &= S^+(t) + 2\alpha\epsilon\left[\sum_{i=0}^{\hat{b}}\sum_{j=0}^{\hat{k}} B_iK_j\right] - \left[\alpha+\delta^++\hat{k}\beta\right]B_{\hat{b}}K_{\hat{k}} \\
\tfrac{d}{dt}B_iK_{\hat{k}} &= 2\alpha(1-\epsilon)\left[\sum_{j=1}^{\hat{k}} B_{i+1}K_j\right] - \left[\alpha+\delta^++\hat{k}\beta\right]B_iK_{\hat{k}} &&\text{for } 0<i<\hat{b} \\
\tfrac{d}{dt}B_0K_{\hat{k}} &= S^-(t) + 2\alpha(1-\epsilon)\left[\sum_{j=0}^{\hat{k}} B_1K_j + B_0K_j\right] - \left[\alpha-\delta^++\hat{k}\beta\right]B_0K_{\hat{k}} \\
\tfrac{d}{dt}B_iK_j &= \hat{k}\beta B_iK_{j+1} - \left[\alpha+\delta^++\hat{k}\beta\right]B_iK_j &&\text{for } 0\leq i\leq\hat{b} \\
&&&\text{and } 0<j<\hat{k} \\
\tfrac{d}{dt}B_iK_0 &= \hat{k}\beta B_iK_1 - \left[\alpha+\delta^-\right]B_iK_0 &&\text{for } 0\leq i\leq\hat{b}.
\end{aligned}
\tag{14}
$$

Here $S^-$ and $S^+$ are respectively the rates of entry of unlabelled and labelled cells into memory from the source population, and $\epsilon$ is the efficiency of uptake of BrdU per division.

## Initial conditions and population size constraints

All cells are unlabelled initially,

$$
B_iK_j(0) = 0 \quad \text{for } 0 < i \leq \hat{b} \text{ and } 0 \leq j \leq \hat{k}.
\tag{15}
$$

Ki67$^{\text{high}}$ cells are assumed to be in a steady-state distribution of stages of loss of Ki67 following mitosis. Empirically, BrdU labelling has no detectable effect on Ki67 expression levels and total numbers of memory cells are constant across the course of the experiments (*Appendix 1—figure 1*). We therefore assume that the total numbers of cells within the Ki67$^{\text{low}}$ and each of the Ki67$^{\text{high}}$ states (irrespective of label content) are all constant over time:

$$
\sum_{i=0}^{\hat{b}} B_iK_0(t) = 1 - \kappa
\tag{16}
$$

$$
\frac{d}{dt}\sum_{i=0}^{\hat{b}} B_iK_j(t) = 0 \quad \text{for } 0 \leq j \leq \hat{k},
\tag{17}
$$

where $\kappa$ is the measured and constant proportion of cells that are Ki67$^{high}$ and we have normalised the total population size to 1. As these relations hold for all $t$, setting $t = 0$ gives two constraints,

$$B_0 K_0(0) = 1 - \kappa \tag{18}$$

$$\sum_{j=1}^{\hat{k}} B_0 K_j(0) = \kappa. \tag{19}$$

The transitions out of each state (*i.e.* death, re-entry into division or progression to the next, lower, state of Ki67 expression) are assumed to obey first order kinetics, so the size of each Ki67$^{high}$ compartment is simply a constant proportion of the one upstream of it. Specifically,

$$
\begin{aligned}
B_0 K_j(0) &= \frac{\hat{k}\beta}{\alpha + \delta^+ + \hat{k}\beta} B_0 K_{j+1}(0) \quad \text{for } 0 < j < \hat{k} \\
B_0 K_0(0) &= \frac{\hat{k}\beta}{\alpha + \delta} B_0 K_1(0).
\end{aligned}
\tag{20}
$$

*Equations 15 and 18–20* specify the initial conditions for all compartments. The constraints of constant pool size and constant Ki67$^{high}$ fraction allow us to solve for two of the free parameters. We choose these to be the rate of entry into division $\alpha$ and the Ki67$^{low}$ loss rate $\delta^-$.

## Fitting the model

When fitting the delabelling curves, we use the conditions at $t = 0$ and propagate the system forward to give the predicted state of the system at the beginning of the delabelling. We then solve and propagate the system forward to give the predicted state of the system at the beginning of the delabelling. We then solve *Equations 14* but with no BrdU uptake ($\epsilon = 0$) and with specified forms for the time-dependence of the labelled and unlabelled source ($S^+(t)$ and $S^-(t)$). Fitting is performed on the timecourses of the two proportions

$$
\begin{aligned}
f^{high}(t) &= \text{Fraction of cells BrdU}^+ \text{ within the Ki67}^{high} \text{ population} \\
f^{low}(t) &= \text{Fraction of cells BrdU}^+ \text{ within the Ki67}^{low} \text{ population.}
\end{aligned}
\tag{21}
$$

## Modelling the BrdU content of the source

We assume that differentiation from naive to memory is linked to cell division and so cells entering from the source are all Ki67$^{high}$. We also assume that during labelling the source is entirely BrdU$^+$. During delabelling we expect there to be a shift from labelled to unlabelled source. We explored a 'delayed-switch' model in which the shift to a BrdU$^-$ source occurs a time $\tau$ post-labelling,

$$
\begin{aligned}
S^-(t) &= \begin{cases} 0 : t < t_{off} + \tau \\ S : t \geq t_{off} + \tau \end{cases} \\
S^+(t) &= S - S^-(t),
\end{aligned}
\tag{22}
$$

where $t_{\text{off}}$ is the time at which the label is withdrawn and $S = S^+ + S^-$ is the total magnitude of the source, assumed to be constant over the course of the labelling experiments. We did not consider a similar delay at the onset of labelling while the source switches from BrdU$^-$ to BrdU$^+$. We assumed that cells divide during the process of differentiation into memory, which means that any such delay could only be caused by inefficient label uptake and so likely occurs on a timescale shorter than $\tau$. We estimated the efficiency of label

uptake to be around 80% for all levels of the source that we explored, in line with other estimates (*De Boer and Perelson, 2013*), and so by parsimony we did not add an additional parameter to describe the early labelling kinetics in the source. We also considered models of a smooth transition in the post-labelling BrdU content of the source, but these gave poorer descriptions of the data.

## Parameter estimation for models of temporal and kinetic heterogeneity

*Equations 14 and 22* describe the labelling and delabelling of a single population of cells with two death rates $\delta^-$ and $\delta^+$, representing temporal heterogeneity. The pool-averaged Ki67$^{high}$ fraction $\kappa$ was determined empirically from the data, and the source $S$ was an input to the model. By eliminating $\alpha$ and $\delta^-$ the model is determined by the unknowns $\beta$, $\epsilon$, $\delta^+$, $\hat{k}$, $\hat{b}$, $\tau$. To fit the models we scanned over sets of the integer values of the numbers of Ki67$^{high}$ and BrdU$^+$ compartments ($\hat{k}$ and $\hat{b}$), and for each pair we estimated the remaining parameters by fitting *Equation 14* to the timecourses of the observables given in *Equation 21* . These quantities were logit-transformed to normalise residuals. Using the maximum likelihood estimates of the variance of the residuals in each of the timecourses, fitting a model of temporal heterogeneity to the BrdU/Ki67 timecourses therefore required maximising the product of the likelihoods of the two timecourses (See Supporting Information S2 in *Hogan et al. (2015)* for a description) over four continuous parameters ($\beta$, $\epsilon$, $\delta^+$, $\tau$) and two discrete ones ($\hat{k}$, $\hat{b}$).

To model multiple subpopulations (kinetic heterogeneity, KH) we used replicates of *Equations 14*. This process increased the number of parameters to be estimated and so we explored a model of two distinct subpopulations only. We therefore had two sets of subpopulation-specific parameters ($\alpha_i$, $\delta_i^-$, $\delta_i^+$, $\kappa_i$, $S_i$) and the global parameters ($\beta$, $\epsilon$, $\hat{k}$, $\hat{b}$, $\tau$). In the simplest KH model we assumed that within each subpopulation Ki67$^{low}$ and Ki67$^{high}$ cells are lost at the same rate; $\delta_i^- = \delta_i^+ \equiv \delta_i$. Both subpopulations were assumed to be in equilibrium, allowing us to eliminate $\alpha_i$ and $\delta_i$ for each. Two further constraints were that the weighted average of the Ki67$^{high}$ fractions within each subpopulation must equal the observed Ki67$^{high}$ fraction in the memory subset, and the weighted average of the influx into each subpopulation must equal the estimated flux of cells into the memory subset.

Consider two kinetically distinct sub-populations, $A$ and $B$, which evolve independently (*i.e.* there is no significant interconversion between $A$ and $B$). If the overall proportion of cells that are Ki67$^{high}$ is $\kappa$,

$$\kappa = q\kappa_A + (1-q)\kappa_B \tag{23}$$
$$S = S_A + S_B \tag{24}$$

where $q$ is the proportion of cells in population $A$. We define $A$ to be the faster subpopulation with the higher Ki67$^{high}$ fraction, so $0 \leq \kappa_B \leq \kappa \leq \kappa_A \leq 1$. In the text (*Table 2* and *Figure 5* ) we show estimates of the fraction of the total source feeding the slow subpopulation,

$$\zeta = \frac{S_B}{S}. \tag{25}$$

In summary, a two-population model of kinetic heterogeneity is parameterised by ($\beta$, $\epsilon$, $\kappa_A$, $\kappa_B$, $\zeta$, $\tau$, $\hat{k}$, $\hat{b}$). The remaining parameters ($\alpha_A$, $\alpha_B$, $\delta_A$, $\delta_B$ , $q$) can be calculated in terms of these unknowns, the overall Ki67$^{high}$ fraction $\kappa$, and the total source $S$. Therefore for each discrete pair ($\hat{k}$, $\hat{b}$) we estimated six continuous parameters ($\beta$, $\epsilon$, $\kappa_A$, $\kappa_B$, $\zeta$, $\tau$). We generated

bootstrap confidence intervals on these parameters by resampling residuals 1500 times and refitting. Uncertainties in the Ki67$^{high}$ fraction $\kappa$ and the source $S$ were propagated into these confidence intervals by simultaneously drawing values of $\kappa$ and $S$ from their empirical (bootstrap) distributions.

Gossel *et al*. eLife 2017;6:e23013. DOI: 10.7554/eLife.23013

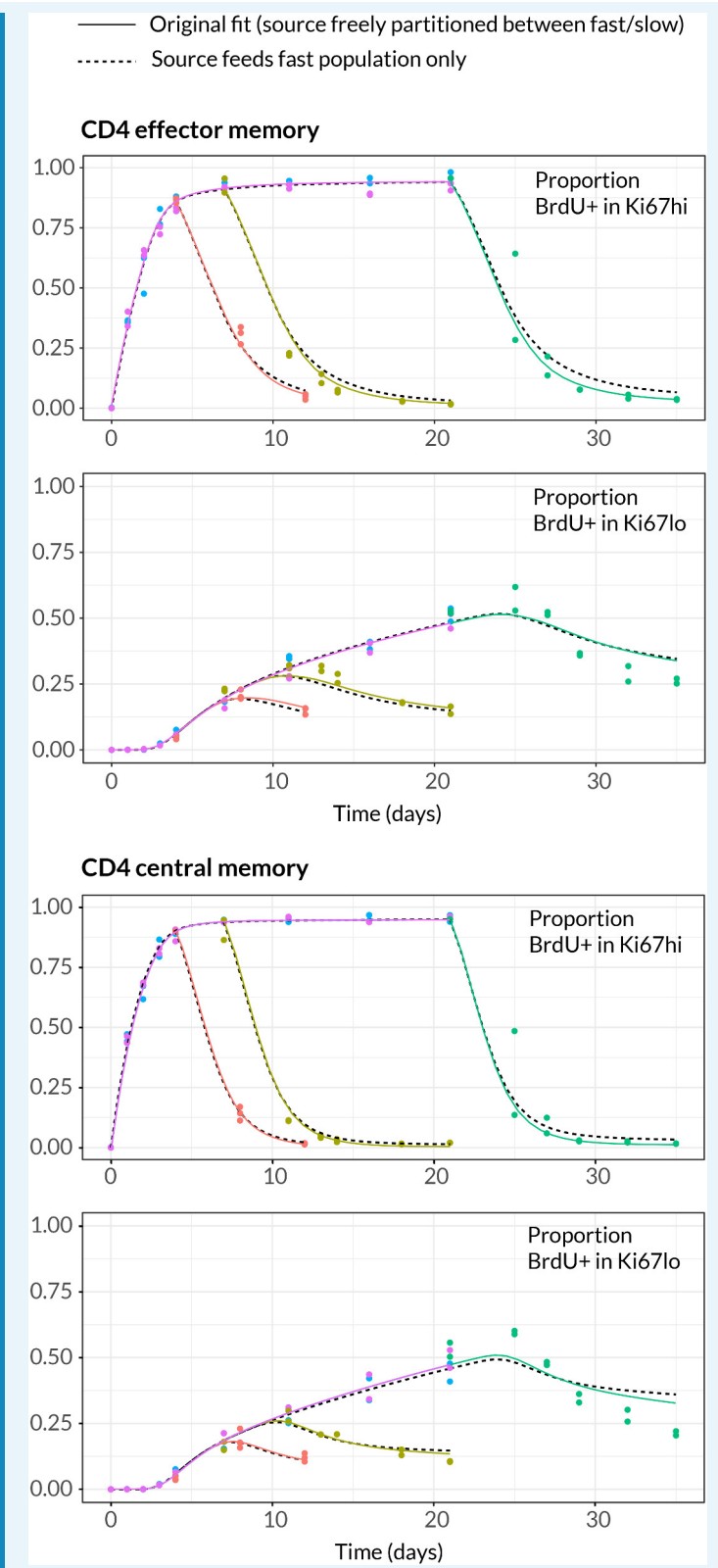

**Appendix 1—figure 2.** Comparing models of BrdU labelling with different partitionings of the influx into memory. We compare the fits using the model presented in the text, in which the allocation of the source into fast/slow populations is a free parameter (solid lines) with a reduced model in which the source is forced to feed the fast subpopulations only (dashed lines).

**Appendix 1—table 3.** Comparing parameter estimates for variants of the model of kinetic heterogeneity used to describe BrdU labelling kinetics. We show point estimates of parameters for the original KH model (source freely allocated to the kinetically distinct subpopulations) and the variant which the source is forced to feed into the fast subpopulation only. The impact is largely on the parameters defining the slow population.

**CD4 effector memory**

| Quantity | Free source | Source feeds fast |
|---|---|---|
| Brdu uptake efficiency (%) | 76 | 79 |
| Mean Ki67 lifetime (days) | 3.28 | 3.27 |
| Population-averaged lifetime (days) | 29 | 39 |
| Population-averaged interdivision time(days) | 88 | 39 |
| Delay before source turns BrdU⁻ post-labelling (days) | 2.52 | 2.43 |
| Source contribution to peripheral production (fraction) | 0.12 | 0.13 |
| Lifetime of fast population (days) | 5.7 | 5.4 |
| Lifetime of slow population (days) | 43 | 58 |
| Interdivision time of fast population (days) | 5.7 | 6.3 |
| Interdivision time of slow population (days) | 138 | 58 |
| Size of fast population (fraction of total) | 0.38 | 0.37 |

**CD4 central memory**

| Quantity | Free source | Source feeds fast |
|---|---|---|
| Brdu uptake efficiency (%) | 77 | 79 |
| Mean Ki67 lifetime (days) | 3.59 | 3.51 |
| Population-averaged lifetime (days) | 21 | 30 |
| Population-averaged interdivision time(days) | 86 | 31 |
| Delay before source turns BrdU⁻ post-labelling (days) | 0.085 | 0.003 |
| Source contribution to peripheral production (fraction) | 0.092 | 0.091 |
| Lifetime of fast population (days) | 3.3 | 3.23.2 |
| Lifetime of slow population (days) | 38 | 57 |
| Interdivision time of fast population (days) | 3.4 | 3.5 |
| Interdivision time of slow population (days) | 167 | 57 |
| Size of fast population (fraction of total) | 0.49 | 0.49 |

