## [Decision Letter]

Thank you for submitting your article "Memory CD4 T cell subsets in mice are kinetically heterogeneous and replenished from naive T cells at high levels" for consideration by *eLife*. Your article has been favorably evaluated by Michel Nussenzweig as the Senior Editor, Rob de Boer as the Reviewing Editor and Reviewer #1, and two peer reviewers: Jose Borghans (Reviewer #2) and Ruy Ribeiro (Reviewer #3).

The reviewers have discussed the reviews with one another and the Reviewing Editor has drafted this decision to help you prepare a revised submission.

Summary:

The main novelty of the study is to include ablation and reconstitution of the hematopoietic stem cell compartment, which allows for the measurement of flow of "new" cells into the memory compartment, i.e. to quantify the source of cells into these compartments. The maintenance of memory T cell populations in non-immunized mice is shown to depend strongly on a tonic influx of naive T cells. In 14 week-old mice 5-10% of the memory T cells are replaced per week by cells from the naive compartment. Both the effector-memory (TEM) and the central memory (CM) T cell compartments consist of at least two subpopulations that are either long-lived (half a year) or short-lived (less than a week).

Important revisions:￼￼

1) It would be helpful to add more (possible) interpretations of the results. For instance, it could be emphasized more that these are "clean" laboratory mice with few memory T cells. The nature of the memory pools in unchallenged mice has always been elusive. If these are cells responding to environmental and food antigens, it becomes much more natural that for such cells there is a tonic flow from naive to memory because these are "persisting" antigens that continue to activate novel naive T cells. The group of Rafi Ahmed (Choo JI 2010) convincingly demonstrate that memory T cells generated during anti-viral immune responses are relatively long-lived, and these cells are maintained in the absence of a flow from naive to memory (because the antigen is not persistent). Summarizing, it would be good to emphasize that this tonic flow could be related to persisting antigens only, and hence be valid for a subset of the memories only.

2) A second example is the fact that about 50% of the population is incumbent in *both* the CM and EM populations (which I first found to be very surprising). However, if we turn things around, and argue that there are incumbent memory T cell clonotypes (that are slow and specific for non-persisting antigens, like Rafi's memory cells), and more activated memory cells (maybe specific for persisting antigens) with a major input from novel naive T cells activated by these antigens, then we only have to argue that each of these two pools is segregated into CM and EM cells. It is then much less coincidental that both CM and EM have about 50% incumbent cells.

3) It would be good to make a better connection between the incumbent cells and the slow sub-compartments in the CM and EM pools. Do you think they are the same (see my reasoning above, arguing that they could be)? How do you think the two subpopulations defined in the paragraph about Ki67 and BrdU relate to the incumbent and replaceable populations of the first half of the manuscript? It is somewhat puzzling is that the cells resistant to displacement are those turning over faster. Do you have any more insight what could be defining this population? How can it be created before 6(?) weeks and then maintained more or less indefinitely?

4) One of the most surprising findings is that central memory and effector memory CD4^+^ T-cells have very similar kinetics. However, this is not discussed appropriately. Did you expect this? Is it not surprising? Are these two subsets maintained by similar processes / mechanisms? Are there really two distinct subsets (I understand the surface markers difference…)?

5) The method of busulfan treatment that is used is said to "leave compartments of committed lineages intact", but Figure 1 shows that the memory pool shortly after BMT is clearly smaller than a few weeks post-BMT. Have the authors also counted cell numbers before busulfan treatment? Is the increase in memory cell numbers and the decline in naive cell numbers completely caused by aging, or could these changes partially reflect (side) effects of busulfan treatment on the peripheral compartments? It is also not clear if the level of DP1 chimerism is changing in time or not. I infer from the text that it is not after a certain time point, but it would be worth showing this in a figure, since this is used to normalize all the data.

6) According to the authors, the model analysis of Figure 2 provides "strong support" for the option that new TEM are recruited directly from the naive compartment rather than from TCM. I find this interesting, but surprising for two reasons. Firstly, by eye the fits of the two models do not look very different and it is even hard to judge which of the models is describing the data better, especially because cell number data tend to be quite noisy. Secondly, the model in which TEM are sourced by the naive pool rather than by TCM contradicts quite some literature concluding that TEM are sourced by TCM (as nicely reviewed by Restifo et al. 2013). The authors should at least discuss what could explain this difference.

7) The most difficult part is to understand the results of Figure 2.

It remains unclear to me why the changes in% replaced with age differ so much between TCM and TEM. Since the changes in chimerism with age (Figure 2) do not differ much between both subsets, I guess it must be the large difference in changes in cell numbers with age that is driving this. It would be very helpful if the authors could explain this in more detail/ more intuitively.

In the same panels, I find it confusing that the "%Total pool replaced" is so similar to the "%Displaceable replaced" for TCM but not for TEM.

The results of Figure 2—figure supplement 1 also confuse me. According to these predictions the TEM pool would consist of nearly 100% incumbent cells at age 100 days. Is that a direct reflection of the low chimerism at day 100 reported in Figure 2? Is the slow increase of the chimerism with age not due to the time it takes for the replaceable cells to be replaced?

The authors conclude that the rate of recruitment into memory from the naive pool varies with age. Can one safely conclude that, or should one in fact perform busulfan treatment at higher ages to find this out?

8) The authors prefer a model with two kinetic populations, although they say that likely there are even more kinetically differentiated populations, but the study does not have power to probe these. One alternative model is to have just one population with a distribution of turnover rates, as proposed in Ganusov et al. PLoS Comp BIol 2010. This model has the same (or approximately the same) number of parameters and perhaps could offer a more biological and parsimonious explanation of the current data.

9) It would be helpful to have an intuitive explanation for the reason the temporal heterogeneity model is not fitting the data well in Figure 4. It would be important to provide some insight into why the temporal heterogeneity model does not work. Currently, when can see that the fits are poorer (both in Figure 4 and by AIC), but do you have any understanding why this is the case from studying the model. In particular, it is a little strange that the fits are so tight, but so poor.

10) It would be interesting to show what the prediction of the model for the fraction of chimerism in the slow and fast turning over populations is. Do you have data on this that you could compare to the model predictions?

11) The modeling methods would benefit from a little more detail in some parts. For instance, in the subsection “M2 Modelling the fluxes between naive, central memory and effector memory subsets” what do you mean by "using exponential-smoother functions", or what is your assumption for γ(t) – you say that you estimate γ, but here the model presented has a γ(t). It is not clear how you obtain equations M2.7 and M2.8. In the last paragraph of the aforementioned subsection, why is 100 days of age special?

12) Equations A1 are hard to follow. You define B^-^K^-^ and B^-^K^+^ but these are not used in the equations. Are these not important here? This nomenclature then reappears in equations A2 and after. What is S_G_ in A9? The last paragraph of the subsection “Parameter estimation for models of temporal and kinetic Heterogeneity” is not completely clear. What do you mean "using a value of the total magnitude of the source chosen randomly from its bootstrap distribution"?

---

## [Author Response]

*Important revisions:*

*1) It would be helpful to add more (possible) interpretations of the results. For instance, it could be emphasized more that these are "clean" laboratory mice with few memory T cells. The nature of the memory pools in unchallenged mice has always been elusive. If these are cells responding to environmental and food antigens, it becomes much more natural that for such cells there is a tonic flow from naive to memory because these are "persisting" antigens that continue to activate novel naive T cells. The group of Rafi Ahmed (Choo JI 2010) convincingly demonstrate that memory T cells generated during anti-viral immune responses are relatively long-lived, and these cells are maintained in the absence of a flow from naive to memory (because the antigen is not persistent). Summarizing, it would be good to emphasize that this tonic flow could be related to persisting antigens only, and hence be valid for a subset of the memories only.*

We agree. Though we did allude to this in the Discussion; we now bring it out more in the second paragraph. We would also argue that much of our understanding of memory turnover (including that deriving from Rafi’s studies) relates to CD8 and not CD4 T cells – and one shouldn’t assume they are similar.

*2) A second example is the fact that about 50% of the population is incumbent in both the CM and EM populations (which I first found to be very surprising). However, if we turn things around, and argue that there are incumbent memory T cell clonotypes (that are slow and specific for non-persisting antigens, like Rafi's memory cells), and more activated memory cells (maybe specific for persisting antigens) with a major input from novel naive T cells activated by these antigens, then we only have to argue that each of these two pools is segregated into CM and EM cells. It is then much less coincidental that both CM and EM have about 50% incumbent cells.*

Yes, this is a nice observation – although we are now showing data (see next point) that suggests that the incumbents are not exclusively slow cells. But basically, yes – one could infer simply that resistant memory has a similar EM/CM composition to that of displaceable memory.

*3) It would be good to make a better connection between the incumbent cells and the slow sub-compartments in the CM and EM pools. Do you think they are the same (see my reasoning above, arguing that they could be)? How do you think the two subpopulations defined in the paragraph about Ki67 and BrdU relate to the incumbent and replaceable populations of the first half of the manuscript?*

This is a good suggestion and we have added some data to make a stronger connection between the two elements. Accumulating sufficient age-matched chimeras to perform a detailed BrdU/Ki67 analysis split over host and donor is currently beyond us (we’re working on it, and it’s for a further study). However, we now include a figure showing Ki67 expression split by donor and host and CD4 EM/CM in chimeras of similar ages to those used in the BrdU analysis (~14 weeks) – see new panel in Figure 5.

It is clear that a larger proportion of donor cells express Ki67 than host. Yet the differences are not as extreme as the fast/slow split we infer from the BrdU analysis – so fast/slow does not simply map to donor/host, and both are likely heterogeneous. So, if chronic stimuli are responsible for driving relatively rapid turnover, and transient stimuli elicit slow dividing memory, perhaps to infection as you suggest, then we see evidence for the both amongst new (donor) and more established (host, possibly incumbent) populations.

*It is somewhat puzzling is that the cells resistant to displacement are those turning over faster. Do you have any more insight what could be defining this population? How can it be created before 6(?) weeks and then maintained more or less indefinitely?*

We did not connect the non-replaced (host) cells in the first part of the analysis to either the fast or slow populations that emerged from the BrdU analysis. Indeed with the new data we now show that host memory cells appear to be slightly enriched for slower cells compared to donor.

Regarding the assumption of long-term stability – it comes from parsimony and resolving power. It was simply not possible to detect any significant growth or decay of this population. We observed the same issue with naive cells in our earlier study. All we are saying is that the attainment of steady peripheral chimerism and steady memory cell numbers is consistent with a model of memory homeostasis that includes a numerically stable, incumbent subset generated pre-BMT.

*4) One of the most surprising findings is that central memory and effector memory CD4^+^ T-cells have very similar kinetics. However, this is not discussed appropriately. Did you expect this? Is it not surprising? Are these two subsets maintained by similar processes / mechanisms? Are there really two distinct subsets (I understand the surface markers difference…)?*

We agree this is surprising, and yes, we think it suggests that in CD4 memory these populations are not as distinct as dogma (arguably driven more by CD8 studies) would have it. We conclude that the CD4 EM/CM split on the basis of L-selectin expression alone doesn’t identify kinetically uniform populations, suggesting that these subsets are also functionally diverse. This diversity, and the EM/CM lineage relationships in CD4 memory, therefore need further analysis. We’ve added text to the Discussion on this point.

*5) The method of busulfan treatment that is used is said to "leave compartments of committed lineages intact", but Figure 1 shows that the memory pool shortly after BMT is clearly smaller than a few weeks post-BMT. Have the authors also counted cell numbers before busulfan treatment? Is the increase in memory cell numbers and the decline in naive cell numbers completely caused by aging, or could these changes partially reflect (side) effects of busulfan treatment on the peripheral compartments? It is also not clear if the level of DP1 chimerism is changing in time or not. I infer from the text that it is not after a certain time point, but it would be worth showing this in a figure, since this is used to normalize all the data.*

Yes, it is important to clarify these control issues. We’ve now reworked Figure 1to show that peripheral compartment sizes across age look normal in chimeras.

Regarding the long-term stability of DP1 chimerism; since sampling is destructive this is hard to test directly, but in Figure 1 in Hogan PNAS 2015 we showed (i) chimerism stabilised within the thymus by 6 weeks post-BMT, (ii) no evidence for a consistent trend in DP1 chimerism across animals out to 52 weeks post-BMT, and (iii) there was no impact of treatment on total thymocyte numbers. We now describe and reference those data.

*6) According to the authors, the model analysis of Figure 2 provides "strong support" for the option that new TEM are recruited directly from the naive compartment rather than from TCM. I find this interesting, but surprising for two reasons. Firstly, by eye the fits of the two models do not look very different and it is even hard to judge which of the models is describing the data better, especially because cell number data tend to be quite noisy. Secondly, the model in which TEM are sourced by the naive pool rather than by TCM contradicts quite some literature concluding that TEM are sourced by TCM (as nicely reviewed by Restifo et al. 2013). The authors should at least discuss what could explain this difference.*

This is a fair point and it’s very useful to clarify this. Note that we did include a section in the Discussion stating that EM are likely sourced by both naive and CM, but that we lack the power to fit a combined model. In response to your comment we’ve removed the statement “strong support” and added the Restifo citation in the Discussion (although it deals predominantly with CD8 and not CD4). We also extended the description of the model discrimination in the Results. Essentially, assessing the fits by eye can be misleading, because the much of the information comes from the upslope in EM chimerism, where there is very little spread in the data (fitting of these fractions was performed on a logit scale to normalise the errors). The naive source model clearly describes these observations better than a CM source.

*7) The most difficult part is to understand the results of Figure 2.*

*It remains unclear to me why the changes in% replaced with age differ so much between TCM and TEM. Since the changes in chimerism with age (Figure 2) do not differ much between both subsets, I guess it must be the large difference in changes in cell numbers with age that is driving this. It would be very helpful if the authors could explain this in more detail/ more intuitively.*

We agree that the analysis of the source required more intuitive explanations, and we have revised appropriately in multiple places. You’re right – this effect is due to different trends in cell numbers with age.

*In the same panels, I find it confusing that the "%Total pool replaced" is so similar to the "%Displaceable replaced" for TCM but not for TEM.*

This is due to EM/CM differences in the estimated sizes of the incumbent populations, although note that these come with considerable confidence intervals (Figure 2—figure supplement 1). For EM, the point estimate of the incumbent population size as a proportion of the pool is initially very high; so that daily replacement as a fraction of the total pool is low, but is close to 100% of the small displaceable subset. For CM, the incumbent fraction is small and so% total pool replaced and% displaceable replaced are more similar.

We’ve added a line to make this clearer.

*The results of Figure 2—figure supplement 1 also confuse me. According to these predictions the TEM pool would consist of nearly 100% incumbent cells at age 100 days. Is that a direct reflection of the low chimerism at day 100 reported in Figure 2?*

Not exactly – the estimate of close to 100% incumbents at 100d arises from a combination of three factors: (i) the fact that the EM pool grows with age, (ii) the assumption that incumbents are stable in numbers, and (iii) donor cells are entering EM at a significant rate – reflected by the high chimerism in the source – yet the EM chimerism remains relatively low. Together these mean that at early timepoints the displaceable EM population must be relatively small.

*Is the slow increase of the chimerism with age not due to the time it takes for the replaceable cells to be replaced?*

No – we infer that the displaceable cells are replaced quickly. The slow shift in chimerism is due to the slow net growth of the donor-rich displaceable populations, while the host incumbent cells remain stable in number.

*The authors conclude that the rate of recruitment into memory from the naive pool varies with age. Can one safely conclude that, or should one in fact perform busulfan treatment at higher ages to find this out?*

This is a great point and we are planning experiments with older chimeras – though these data will not be available for several months. This is the next study.

Basically, yes – in the resistant memory model, we predict that rate of influx in terms of absolute #s of cells per unit time declines in tandem with naive T cell numbers.

Our core result relates to the magnitude of the source at around 14 weeks of age. The data are conclusive on this point, since the upslopes in chimerism at that age are very clearly defined. The relative sparsity of data later on- and in particular, uncertainty in the trends in the sizes of the EM and CM pools – mean that the predictions in Figure 2 are more speculative, as reflected in the confidence intervals.

*8) The authors prefer a model with two kinetic populations, although they say that likely there are even more kinetically differentiated populations, but the study does not have power to probe these. One alternative model is to have just one population with a distribution of turnover rates, as proposed in Ganusov et al. PLoS Comp BIol 2010. This model has the same (or approximately the same) number of parameters and perhaps could offer a more biological and parsimonious explanation of the current data.*

We like that model and did indeed consider it, but faced the issue of parameterising how the source is then distributed continuously across the distribution of subpopulations, which substantially increases the number of parameters. We acknowledge that there is likely richer substructure, maybe even dynamic, and have extended the discussion on this point (and also include Vitaly’s ref.). But we believe our analysis captures the key issue that there is kinetic heterogeneity at multiple levels within CD4 memory subsets, and that one model of pure TH is ruled out.

*9) It would be helpful to have an intuitive explanation for the reason the temporal heterogeneity model is not fitting the data well in Figure 4. It would be important to provide some insight into why the temporal heterogeneity model does not work. Currently, when can see that the fits are poorer (both in Figure 4 and by AIC), but do you have any understanding why this is the case from studying the model.*

Yes, this is important, and we were puzzled too. Essentially we believe the answer is that in the TH model, the kinetic of accumulation of the BrdU^+^Ki67^-^ population is heavily influenced by that of their BrdU^+^Ki67^+^ precursors – there is both a simple proportional flow from K^+^ to K^-^, and the empirical constraint of the global Ki67 fraction tightly couples the K^+^ and K^-^ populations. There is also a tight constraint on the Ki67 lifetime, defined by the obvious delay before BrdU^+^Ki67^-^ cells first appear. There is therefore very little wiggle room with which to fit both the time courses well. In the KH model, we allow the kinetics of the B^+^K^-^ and B^+^K^+^ to be more decoupled, since they are enriched for the slow and fast populations respectively, and the two interact only weakly through the constraint of the total Ki67 fraction being constant.

We’ve tried to paraphrase this in the caption to Figure 4.

*In particular, it is a little strange that the fits are so tight, but so poor.*

This is an interesting point and is an artefact of bootstrapping residuals (because we’re working with time series) and not the data themselves. The empirical bootstrap distributions of parameters are used to generate the confidence envelopes on the fits; but strictly this procedure is only precisely valid if the underlying model is correct. An illustration is in Figure 8: generate a noisy sine wave and fit a straight line to it. Then resample residuals and generate a confidence envelope – which is clearly not very useful.

Author response image 1.**DOI:**
http://dx.doi.org/10.7554/eLife.23013.026

If one were to use the ‘true’ bootstrapping of the observations themselves, these envelopes would be larger, and would give a truer reflection of the sensitivity of parameter estimates to noise in the data. This is what the envelopes in the KH fits show more meaningfully, since it is clearly the better model.

*10) It would be interesting to show what the prediction of the model for the fraction of chimerism in the slow and fast turning over populations is. Do you have data on this that you could compare to the model predictions?*

We’re not connecting the fast/slow compartments to host/donor, or modelling them as such. However, we’ve included the breakdown of host/donor Ki67 expression (Figure 5) to show that both host and donor cells contain fast and slow populations.

*11) The modeling methods would benefit from a little more detail in some parts. For instance, in the subsection “M2 Modelling the fluxes between naive, central memory and effector memory subsets” what do you mean by "using exponential-smoother functions", or what is your assumption for γ(t) – you say that you estimate γ, but here the model presented has a γ(t). It is not clear how you obtain equations M2.7 and M2.8. In the last paragraph of the aforementioned subsection, why is 100 days of age special?*

We’ve addressed these points in a revised and expanded Methods, and have included the parameter estimates in an Appendix. Quoting 100 days of age without context was misleading – this just happens to be the age of the animals where the early upslope in peripheral chimerism is apparent, and clearly defined – and so is the age at which we have most confidence in the source term.

*12) Equations A1 are hard to follow. You define B^-^K^-^ and B^-^K^+^ but these are not used in the equations. Are these not important here? This nomenclature then reappears in equations A2 and after. What is S_G_ in A9?*

Yes, thank you – old text had crept into the Methods and so notation was inconsistent. We’ve overhauled this section and cleaned up the notation. S_G_ was simply the total magnitude of the source. Now defined as “S” for simplicity.

*The last paragraph of the subsection “Parameter estimation for models of temporal and kinetic Heterogeneity” is not completely clear. What do you mean "using a value of the total magnitude of the source chosen randomly from its bootstrap distribution"?*

We tried to clarify this. We are describing how we propagate multiple sources of uncertainty into the confidence intervals on the parameters inferred from the BrdU timecourses.